# Pulsed stimuli enable p53 phase resetting to synchronize single cells and modulate cell fate

Harish Venkatachalapathy [1,2], Samuel Dallon [3], Zhilin Yang[4], Samira M Azarin[2], Casim A Sarkar [3]✉ & Eric Batchelor [1]✉

## Abstract

**Oscillatory p53 expression occurs in individual cells responding to DNA breaks. While the majority of cells exhibit the same qualitative response, quantitative features of the oscillations (e.g., amplitude or period) can be highly variable between cells, generating heterogeneity in downstream cell fate responses. Since heterogeneity can be detrimental to therapies based on DNA damage, methods to induce synchronization of p53 oscillations across cells in a population have the potential to generate more predictable responses to DNA-damaging treatments. Using mathematical modeling and time-lapse microscopy, we demonstrated that p53 oscillations can be synchronized through the phenomenon of phase resetting. Surprisingly, p53 oscillations were synchronized over a wider range of damage-induction frequencies than predicted computationally. Recapitulating the range of synchronizing frequencies required, non-intuitively, a less robust oscillator. We showed that p53 phase resetting altered the expression of downstream targets responsible for cell fate depending on target mRNA stability. This study demonstrates that p53 oscillations can be phase reset and highlights the potential of driving p53 dynamics to reduce cellular variability and synchronize cell fate responses to DNA damage.**

**Keywords** p53; Oscillations; Single-cell Heterogeneity; Dynamical Systems; Phase Resetting
**Subject Categories** Computational Biology; DNA Replication, Recombination & Repair

## Introduction

Biomolecular oscillations are observed in a variety of cellular processes such as MAPK signaling (Albeck et al, 2013), cell division (Gerard and Goldbeter, 2009), inflammation (Kellogg and Tay, 2015), and DNA damage (Batchelor et al, 2008). Dynamical features of these oscillations, such as amplitude and frequency, encode information about a stimulus and determine the cellular response (Cheong and Levchenko, 2010). However, biological noise can cause variations in both these features, leading to cell-to-cell variations in the downstream response (Qiao et al, 2022; Reyes et al, 2018; Albeck et al, 2013). Synchronization to an external stimulus has evolved as a strategy to facilitate a more uniform population response in the face of such variability (Jiménez et al, 2022b). This synchrony in cellular response can be crucial for cells in a multicellular organism to respond coherently to regulatory cues. Notably, mechanisms of entrainment of circadian oscillations drive synchronous periodic behavior in response to 24-h light-dark cycles across multiple systems within an organism, allowing for coherent metabolic and hormonal dynamics (Reppert and Weaver, 2002; Roenneberg et al, 2003; Aton and Herzog, 2005). Similarly, oscillations in glycolytic networks (Gustavsson et al, 2015) and NF-κB (Kellogg and Tay, 2015; Gupta et al, 2016) have been shown to respond synchronously under periodic stimuli. This is also typically bolstered by cell–cell communication in the form of paracrine or cell–cell junction-mediated signals (Roenneberg et al, 2003; Gupta et al, 2016; Reppert and Weaver, 2002). However, it is unclear whether this extends to oscillatory contexts where the cellular response is intrinsic to a single cell and such coherence in the response of single cells is not required for effective function.

One such context is oscillations in the p53 transcription factor that occur in response to double-stranded DNA breaks (DSBs) induced by stressors such as endogenous replicative stress and γ-irradiation. Here, p53 levels increase and decrease in an oscillatory pattern with an average period of ~5.5 h in mammalian cells and drive DNA damage responses, including DNA repair (Geva-Zatorsky et al, 2006; Stewart-Ornstein et al, 2017). There is significant variability in pulse frequency between individual cells as well as time between pulses in an individual cell (Batchelor et al, 2009), potentially due to the intrinsic lack of synchronizing cues in this context in the form of cell–cell communication or a periodic external driver. However, the number of DSBs continues to decrease exponentially in the overall population, apparently robust to these fluctuations (Mönke et al, 2017; Loewer et al, 2013). Further, despite cell-to-cell variations in p53 dynamics driving heterogeneous tumor cell behavior under cancer therapies (Batchelor and Loewer, 2017; Friedel and Loewer, 2022; Paek et al, 2016; Xie et al, 2022; Yang et al, 2018; Wu et al, 2017), the potential to

---

[1]Department of Integrative Biology and Physiology, University of Minnesota, Minneapolis, MN 55455, USA. [2]Department of Chemical Engineering and Materials Science, University of Minnesota, Minneapolis, MN 55455, USA. [3]Department of Biomedical Engineering, University of Minnesota, Minneapolis, MN 55455, USA. [4]Laboratory of Cell Biology, National Cancer Institute, National Institutes of Health, Bethesda, MD 20892, USA. ✉E-mail: csarkar@umn.edu; ebatchel@umn.edu

synchronize these dynamics, to our knowledge, remains unexplored. Therefore, we sought to determine if periodic DSB induction could improve synchrony in p53 oscillations.

Using a mathematical model of the p53 response to neocarzinostatin (NCS)-induced DSBs developed by Mönke et al (Mönke et al, 2017), we predicted the existence of synchronization in an interval around the natural p53 pulsing period. We validated the prediction experimentally using live-cell fluorescence microscopy and single-cell tracking, demonstrating increased synchronization of p53 dynamics in response to a periodic damage signal. Surprisingly, we experimentally observed synchronization across a wider range of stimuli periods than was computationally predicted. Using parameter sensitivity analysis, we found non-intuitively that the increased synchronization range required a modified p53 DSB response model with a weaker oscillator that would be more susceptible to biological noise. We also predicted computationally and validated experimentally that the synchronization of p53 dynamics is an example of the phenomenon of phase resetting, in which the period of oscillation is restored to the natural period relatively rapidly upon removal of the external oscillatory driving stimulus. Finally, we showed that the change in p53 pulsing period due to phase resetting translates to alteration in downstream target dynamics for cell fate regulatory genes, and such frequency modulation can result in differential cell fate specification.

## Results

### The p53 response can be synchronized through periodic DSB induction

To identify whether synchronization of p53 pulses through periodic DSB induction was theoretically feasible (Fig. 1A), we used a mathematical model of the p53 DSB response calibrated using MCF-7 breast cancer cells (Mönke et al, 2017) (Fig. 1B) to simulate repeated DSB induction through the radiomimetic drug NCS and analyzed the subsequent p53 response. From deterministic simulations using an NCS-induced birth rate of $b_s = 200$ breaks/h, we found a small region around the natural frequency of the system ($\sim 1/5.5\,h^{-1}$) where p53 had the same frequency as the input DSB waveform (phase-locked) (Fig. 1C). We validated this by calculating a synchronization score (Gupta et al, 2016), which is defined as the fraction of total power on the Fourier spectrum that corresponds to the input frequency and its harmonics. Since a synchronized system would have the same frequency as the input, its corresponding frequency spectrum is expected to show high power at the input frequency and harmonics and low power elsewhere, leading to a synchronization score close to 1. Indeed, this score was close to 1 for the regions in which the system was phase-locked, further validating our hypothesis that p53 can be synchronized to repeated DSB induction (Fig. 1C). This did not change significantly as a function of $b_s$ over a range of 50–1000 breaks/h (Fig. EV1A). In addition, under noisy DSB dynamics, the synchronization score of an ensemble of 1000 simulations per input frequency closely followed that of the deterministic system (Fig. 1C), implying DSB stochasticity was likely not a significant source of variability in synchronization under the conditions analyzed.

To validate the computationally predicted conditions of synchronization, we quantified p53 dynamics in p53-mVenus expressing MCF-7 cells that were treated with either a single dose of 400 ng/mL NCS or two doses spaced 5.5 h apart (Fig. 1D). While the second peak of p53 showed a reduction in amplitude in both dosing conditions, as has been previously observed and attributed to differences in MDM2 levels at the basal state versus in the oscillatory state[4], the second pulse amplitude was more comparable to the first in the double damage dose condition (Fig. 1D). For analysis of the effects on p53 pulse timing, rather than calculations of phase-locking to the external stimulus, which occurred on a shorter time-scale experimentally than was possible computationally, we focused on quantifying the changes in the pulse timing variability between cells. Both single-dose and double-dose conditions showed a synchronized first pulse of p53 accumulation after treatment, with no statistically significant differences in peak timing distributions (Fig. 1E). However, the cells that received a single dose of NCS lost synchrony by the second p53 pulse, while cells that received a second dose of NCS at 5.5 h had a significantly narrower second peak timing distribution ($P < 0.01$) (Fig. 1E). This was further validated by the smaller increases in the median absolute deviation (MAD) of second peak timing over first peak timing (single-dose median: 2.41, double-dose median: 1.33; $P < 0.05$) (Fig. 1F). Overall, cells that had been treated twice with NCS better retained p53 peak timing synchrony.

### p53 pulses can be synchronized over a wider range of stimulus periods than predicted

To experimentally validate the computationally predicted range of stimulus periods across which synchronization is possible, we treated the MCF-7 p53-mVenus cell line with either a single dose of 400 ng/mL of NCS or two doses spaced 3.0–7.0 h apart, with a resolution of 0.5 h (Fig. 2A). In contrast to model predictions of a narrow synchronization range of 5.25 h to 5.65 h, we observed statistically significant increases in synchrony in conditions with two doses spaced 4.0 h to 5.5 h apart when compared to a single dose (Fig. 2B). We did not observe similar synchronization in the 3.0, 3.5, 6.0, 6.5, and 7.0 h conditions. Interestingly, the 3.0 h and 3.5 h conditions did not exhibit a synchronous first pulse as typically expected. Instead, we observed cells with either two small pulses spaced close together or one large pulse spanning the approximate duration of the small pulses, thereby exacerbating heterogeneity in the p53 response in comparison to a single dose (Fig. 2A). In contrast, the 6.0, 6.5, and 7.0 h conditions exhibited a synchronized first pulse similar to a single dose of NCS but were followed by a less synchronized second pulse suggesting loss of synchronization.

Surprisingly, inclusion of stochastic mRNA dynamics in computational simulations, a feature shown to widen the synchronization range in other systems (Kellogg and Tay, 2015; Gupta et al, 2016; Kumpost et al, 2023), did not improve synchronization in our p53 model (Fig. EV1B–D). Taken together, these results indicate that the p53 DSB response can be synchronized over a greater range of stimulus periods than predicted and that the improved synchronization is likely not driven by intrinsic noise.

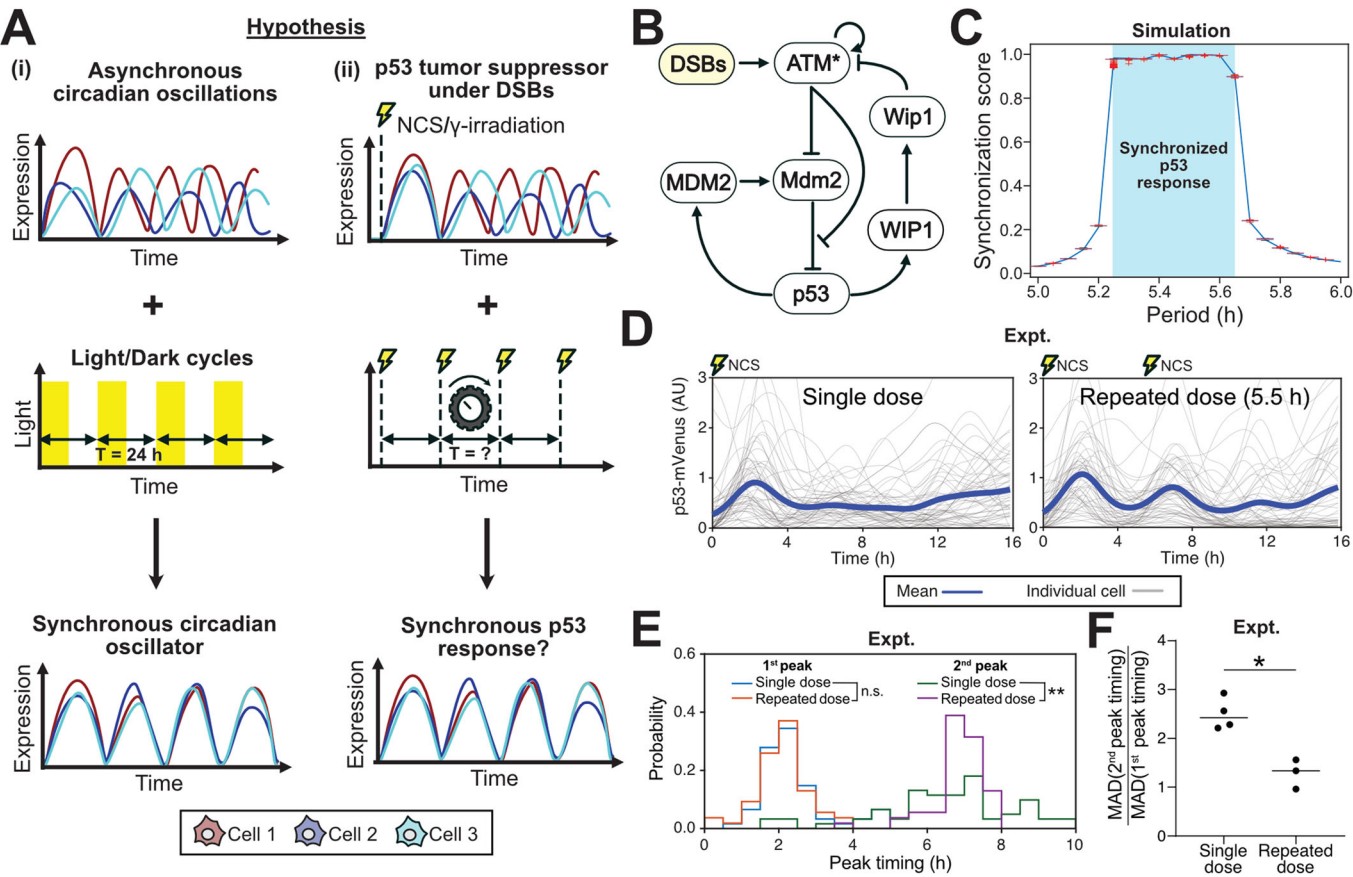

**Figure 1. p53 DSB response can be synchronized.**

(A) (i) Asynchronous oscillations in expression levels of a circadian oscillator are synchronized under periodic light/day cycles. (ii) We hypothesized that asynchronous p53 oscillations under DSBs could be similarly synchronized under periodic DSB induction. (B) Diagram of the p53 regulatory network under DSBs. (C) Synchronization score of deterministic simulations driven by either deterministic (blue line) or stochastic (box plots, 1000 realizations) DSB dynamics. The phase-locked region in the deterministic system is indicated in cyan. (D) Single-cell traces (gray) and mean (dark blue) p53-mVenus transgene expression under a single dose of NCS (400 ng/mL) or two doses spaced 5.5 h apart. (E) Probability distributions of first and second peak timing. (*$P < 0.05$; **$P < 0.01$; Kolmogorov–Smirnov test). (F) Ratio of median absolute deviation (MAD) in peak timing of the second peak to the first, measured across four and three biological replicates for the single-dose and double-dose conditions, respectively. Each biological replicate consists of at least 45 individually tracked cells (*$P < 0.05$; Statistical analysis is ANOVA followed by multiple comparisons that were performed on the dataset presented in Fig. 2. Exact $P$ values provided in Table EV2).

## p53 pulse synchronization arises through phase resetting

Synchronization of oscillators can occur through distinct mechanisms. Two general mechanisms, entrainment and repeated phase resetting, depend on an external periodic stimulus to establish a new period of the oscillators, as we observed for p53 expression. However, the mechanisms differ in how quickly the oscillators revert from the period of the driving stimulus upon removal of the stimulus, with entrained systems persisting in oscillations at the entraining frequency for one or more periods while phase reset systems revert to their natural frequency immediately (Zambrano et al, 2016).

To determine the mechanism of synchronization of the p53 oscillator, we first used our model to predict the duration of time that p53 oscillations would remain at an externally driven frequency upon removal of the external driving stimulus. We simulated a driving stimulus period of 5.3 h, a relatively modest change from the natural p53 oscillation period of ~5.5 h. The model indicated that the p53 period would return to the natural period

almost immediately upon removal of the external stimulus (Fig. EV2A), thus predicting that p53 exhibits phase resetting rather than entrainment.

To validate this prediction experimentally, for the double external dosing experiments in Fig. 2 over the range of 4.0–5.5 h externally driven periods, we compared the timing between the first and second p53 peaks (the first interpeak interval, during the externally driven p53 response) and the second and third peaks (the second interpeak interval, after the external stimulation is halted) (Fig. EV2B). We found that the second interpeak interval returns to the natural peak period of 5.5 h, indicating that as predicted the system returns rapidly to the natural frequency upon removal of external stimulus (Fig. EV2B), along with a restoration of the variability of peak timings (Fig. EV2C). To further validate that the system undergoes phase resetting rather than entrainment, we quantified longer-term p53 dynamics upon removal of externally driven stimulation for a greater number of repeated doses at an interval of 4.0 h or 5.5 h (Fig. EV2D). In all cases, upon cessation of the repeated doses, p53 dynamics rapidly reverted to the natural

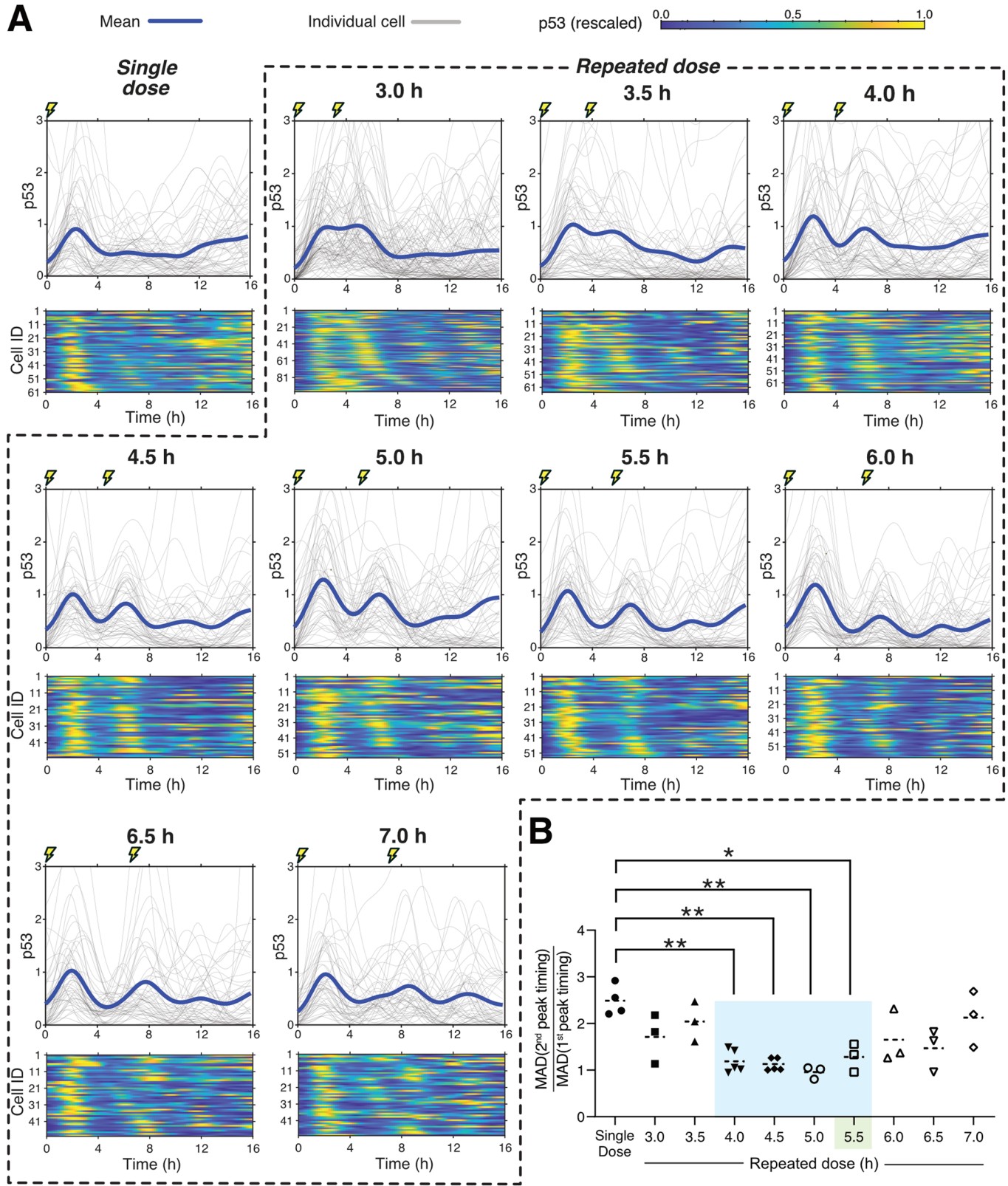

**Figure 2.  p53 is experimentally synchronizable across a wider range of stimulus frequencies than computationally predicted by phase-locking.**

(A) Single-cell traces (gray) and mean (dark blue) line plots and the corresponding per-cell rescaled heatmaps of p53-mVenus expression under a single dose of NCS (400 ng/mL) or two doses spaced 3.0 h to 7.0 h apart. Lightning marks indicate NCS addition. Traces are from a single replicate representative of a minimum of three biological replicates; each biological replicate consists of at least 45 individually tracked cells. (B) Fold change in median absolute deviation (MAD) in peak timing from the first peak to the second, measured across a minimum of three biological replicates (*$P < 0.05$; **$P < 0.01$; ANOVA followed by multiple comparisons. Exact $P$ values provided in Table EV2). Single-dose and 5.5 h double-dose data same as Fig. 1.

oscillatory response in pulse timing and variability (Fig. EV2D,E). Taken together, these results suggest that the p53 response to repeated DNA DSBs behaves as a phase-resetting synchronization system.

## Parameter sensitivity analysis identifies oscillation strength as a predictor of synchronization

Given that intrinsic noise alone was not explanatory of the synchronization range observed experimentally, we performed a parameter sensitivity analysis to examine whether changes in system parameters could explain the experimentally observed range (Fig. 3A). In brief, we varied each parameter one at a time and subjected the system to a constant DSB input. If the resulting output was oscillatory, we identified the period of this system and rescaled the set of differential equations by dividing the right-hand side by the period of the perturbed system ($T_{natural}$) and multiplying it by the period of the original system ($T_0$). This makes the period of the perturbed system equal to the period of the original system. We then stimulated the system with DSB waveforms of periods from ($T_0 - 2$) h to ($T_0 + 2$) h and checked whether the resulting p53 output period was the same as the input period to identify a window of synchronization. To examine sensitivity, we quantified the effect of each parameter on the width of this window of synchronization.

While we found that no single parameter exhibited characteristics that could identify the exact biochemical mechanism behind the increased synchrony observed experimentally, we identified global patterns that could lead to improved phase-resetting behavior. Specifically, we observed two classes of behavior based on whether the range of input frequencies conferring synchronization changed monotonically or nonmonotonically with the parameter. The monotonic class consisted of parameters associated with the p53-Mdm2 loop (e.g., MDM2 mRNA degradation rate constant and Mdm2-mediated p53 degradation rate constant). The non-monotonic class consisted of parameters associated with the p53-Wip1-ATM* interactions (e.g., ATM autophosphorylation rate constant) (Fig. 3B,C). While some parameters had a stronger effect on synchronization than others (Fig. EV3), the most striking feature was a sharp increase in synchronization range at the cusp of the transition to a non-oscillatory regime in some cases, implying that less robust oscillatory behavior improved synchronization. That is, the range of frequencies for which the oscillator could be synchronized in response to an external periodic stimulus was largest in parameter regimes where small changes in parameters (such as those arising from extrinsic noise, a part of natural biological variability) led to the loss of oscillatory behavior. While it is non-intuitive that increased synchronization correlated with less robust oscillations, we hypothesized that an endogenous oscillator with a weaker limit cycle attractor would be more susceptible to

perturbations and thereby more readily synchronized to an external cue. To validate this, we used a trajectory-based energy landscape methodology (Venkatachalapathy et al, 2021, 2024) to generate the underlying energy landscapes for the basal system and compared it to an instance of a more synchronizable system (1.67 times the basal ATM autophosphorylation rate constant). This methodology uses simulated protein abundances sampled from simulated dynamic trajectories to construct a probability distribution ($P$) that is then converted into pseudopotential energy, $U = -\ln(P)$. This allows us to generate a potential energy surface that is reflective of the corresponding probability distribution of system states. We expected oscillators with weaker (less rigid) limit cycle attractors to have a shallower stable steady-state valley that is reflective of higher susceptibility to external perturbations while oscillators with stronger (more rigid) limit cycle attractors would have a deeper valley (i.e., be less affected by external forces) (Fig. 3D). By comparing the two landscapes, we found that the valley around the mean system path in the more readily synchronized system was wider and had a gentler gradient around it (Fig. 3E). This was further corroborated by examining the potential across a line connecting the origin and the inner maximum within the circular steady-state valley. This showed higher stable steady-state potentials and lower slopes around the states in the landscape of the more readily synchronized system, two characteristics that suggest greater responsiveness to external cues. Taken together, the results validate our hypothesis that a less rigid oscillatory system is more affected by external perturbations and thus is more easily synchronized and that the p53 DSB response needs to be underpinned by such a weak oscillator to recapitulate our experimental observations.

## Modulating p53 dynamics through phase resetting changes downstream mRNA target dynamics in a half-life-dependent manner

As p53 dynamics control cell fate in response to DSBs, we hypothesized that changes in p53 dynamics due to phase resetting alter the downstream response. We validated this by examining the expression of the cell cycle inhibitor p21, a canonical p53 downstream target. Comparing the mean p53 expression level across single cells for NCS stimulus periods of 4.0 h and 5.5 h, we observed a shift in the peak timing of the second p53 pulse (Fig. 4A, left). Using a mathematical model of p21 expression (Reyes et al, 2018), we predicted that these p53 dynamics result in higher expression of p21 at the population level between 5.0 h and 8.0 h in the NCS 4.0 h condition, along with an earlier second peak (Fig. 4A, right). To validate this, we quantified p21 protein dynamics in MCF-7 cells tagged with mVenus at the p21 C-terminal genomic locus under different NCS treatment intervals. We only considered cells that had a minimum normalized expression of $10^{0.5}$ units at

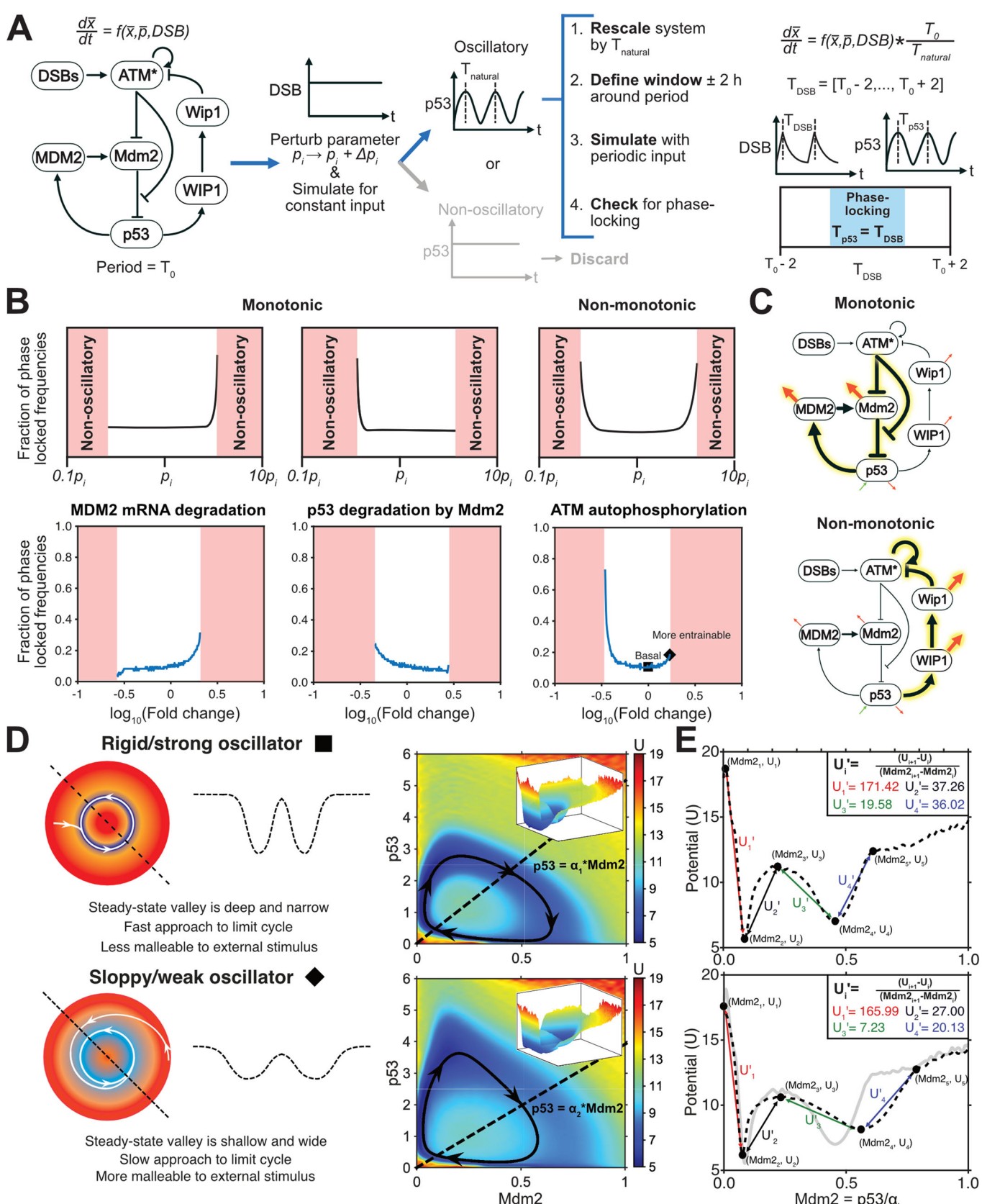

Figure 3. Parameter sensitivity analysis identifies proximity to non-oscillatory regimes and limit cycle rigidity as a positive predictor of phase-locking range.

(A) Schematic depicting the parameter sensitivity analysis workflow. (B) Depictions of the two types of phase-locking range responses to parameter changes in the system with examples of each type. (C) Schematic of the p53 DSB response model with specific interactions that exhibit monotonic or non-monotonic responses highlighted and enlarged in size. Black arrows depict interactions while green and red arrows represent basal production and degradation, respectively. (D) Graphical depictions and the underlying energy landscapes of a more rigid oscillator (basal system; black square in ATM autophosphorylation rate diagram in (B)) and less rigid oscillator (ATM autophosphorylation rate changed to 1.67 times the basal value; black diamond in ATM autophosphorylation rate diagram). Rigidity refers to the strength of the limit cycle attractor. The black line represents the deterministic path (limit of zero noise) and arrows represent direction of movement. The dashed line represents the line passing through the origin and the inner maximum of the steady-state valley described by $Mdm2 = \alpha_i{}^*p53$ where $\alpha_1 = 4.76$ and $\alpha_2 = 4.24$. Energy landscapes generated using p53 DSB model with stochastic DSB and mRNA dynamics and system size of 100. (E) Slice of energy landscapes corresponding to (D) along the dashed line depicting the change in the rigidity of the steady state. Gray line in the bottom panel corresponds to the more rigid landscape (dashed line in the top panel).

$t < 3.0$ h to filter out cell cycle-dependent delays in p21 upregulation (Sheng et al, 2019). Consistent with the model, we found higher mean p21 expression in the NCS 4.0 h condition between 5.0 h to 8.0 h compared to the NCS 5.5 h condition (Fig. 4B). Further, we found that the second peak timing of p21 expression was later for cells administered a second dose of NCS at 5.5 h compared with 4.0 h ($P < 0.0001$; Wilcoxon rank-sum test). No differences were observed in the first peak timings between the two conditions (Fig. 4C). Finally, based on a study by Reyes et al (Reyes et al, 2018), we predicted that lower expression of p21 in the NCS 5.5 h condition that we had observed from 5.0 h to 8.0 h compared to the NCS 4.0 h condition can result in stochastic transitions to a low p21 state during this time, characteristic of escape from cell cycle arrest. Indeed, while there were cells that reverted to a low p21 state in both the NCS 4.0 h and NCS 5.5 h conditions (magenta traces, Figs. 4D and EV4A,B), escape from this high-p21 state between 5.0 h to 8.0 h occured only in the NCS 5.5 h condition (green traces, Figs. 4D and EV4A,B).

Given the rarity of these escaper cells, we extended the duration of phase resetting by increasing the number of NCS doses from two to three and performed targeted quantification of escaper cells in this dataset (Fig. 4E). By examining the duration from 4.0 h to 12.0 h, when both oscillators are considered synchronized, we found escaper cells in both the 4.0 h and 5.5 h treatment conditions during the synchronized period (Fig. 4F). This was further validated by a simultaneous increase in Cdk2 activity in these cells (Fig. EV4C). However, there was a ∼ 4.5-fold increase in the percentage of cells that escaped the high-p21/low Cdk2 state in the 5.5 h regimen compared to the 4.0 h regimen (Figs. 4G and EV4D). This implies that while the rare subpopulation of escaper cells could be found in both conditions, they were much more likely to arise if the input period was 5.5 h compared to 4.0 h.

Since p53 regulates many other target genes in addition to p21 with diverse cellular outcomes, we next sought to determine whether there were effects on the expression of other target genes. Our previous work showed that p53 targets have distinct mRNA expression dynamics depending on whether their mRNA half-lives are faster or slower than the period of p53 oscillations ((Porter et al, 2016), Fig. 5A). We first analyzed by qPCR mRNA expression dynamics for select p53 target genes with a range of mRNA stabilities.

We observed a qualitative increase in mRNA expression of not only *CDKN1A* (p21 mRNA) but also other p53 targets (*GADD45A, TRIAP1,* and *FAS*) between 5.0 and 8.0 h under administration of a second dose of NCS at 4.0 h compared with 5.5 h (Figs. 5B and EV5). However, only *GADD45A* showed a statistically significant difference with $P$ value thresholds of 0.1 and 0.05. This

may be due to differences in mRNA stability—*GADD45A* is highly unstable with an mRNA decay rate constant of $0.599\,h^{-1}$ and follows p53 dynamics more closely in comparison to the more stable species *CDKN1A, TRIAP1,* and *FAS* that have mRNA decay rate constants of 0.248, 0.212, and $0.050\,h^{-1}$, respectively (Porter et al, 2016). Therefore, any changes in p53 dynamics would be better reflected in *GADD45A* dynamics as opposed to more stable downstream mRNA targets. Indeed, by modeling mRNA transcription using p53 dynamics from Fig. 4A as an input, we predicted that less stable mRNA targets show greater differences in expression between the 4.0 h and 5.5 h NCS interval treatments (Fig. 5C). This was experimentally validated by performing qRT-PCR on other similarly unstable targets—*BTG2, PUMA,* and *PMAIP1* (mRNA degradation rate constants of 0.844, 0.730, and $0.558\,h^{-1}$ respectively, much higher than the p53 pulse frequency of $\sim0.18\,h^{-1}$) (Porter et al, 2016). We detected a statistically significant increase in *BTG2* and *PMAIP1* in the 4.0 h condition compared with the 5.5 h condition with a $P$ value threshold of 0.05; the increase in *PUMA* was associated with a $P$ value of 0.051 (Fig. 5D). Thus, phase resetting of the single-cell p53 response can modulate downstream target expression in a target stability-dependent manner with a bias towards genes that are less stable.

## Discussion

Periodic external cues can synchronize biological rhythms across cells. This is particularly important for driving a coherent response in metabolic regulation as well as paracrine/endocrine signaling in multicellular organisms (Jiménez et al, 2022a; Gupta et al, 2016; Reppert and Weaver, 2002; Roenneberg et al, 2003). In this study, we demonstrated that this phenomenon also extends to phase resetting of the p53 oscillatory response to DSBs, a context where cell-to-cell coherence does not underpin an effective cellular response as measured by successful DNA repair. By periodically inducing DSBs through NCS (a small molecule γ-irradiation analog), cell-to-cell heterogeneity in the period of p53 pulses was reduced significantly compared to a single initial dose. Interestingly, the range of periods over which synchronization occurred (>1.5 h) (Fig. 2) was significantly larger than computationally predicted (∼0.5 h) even in conditions of high DNA damage (Figs. 1C and EV1). Further, in contrast to other systems where intrinsic noise widens the synchronization range (Kellogg and Tay, 2015; Gupta et al, 2016; Kumpost et al, 2023), there was no such intrinsic noise-driven range widening in the p53 system (Fig. EV3). Instead, using parameter sensitivity analysis, we found that the

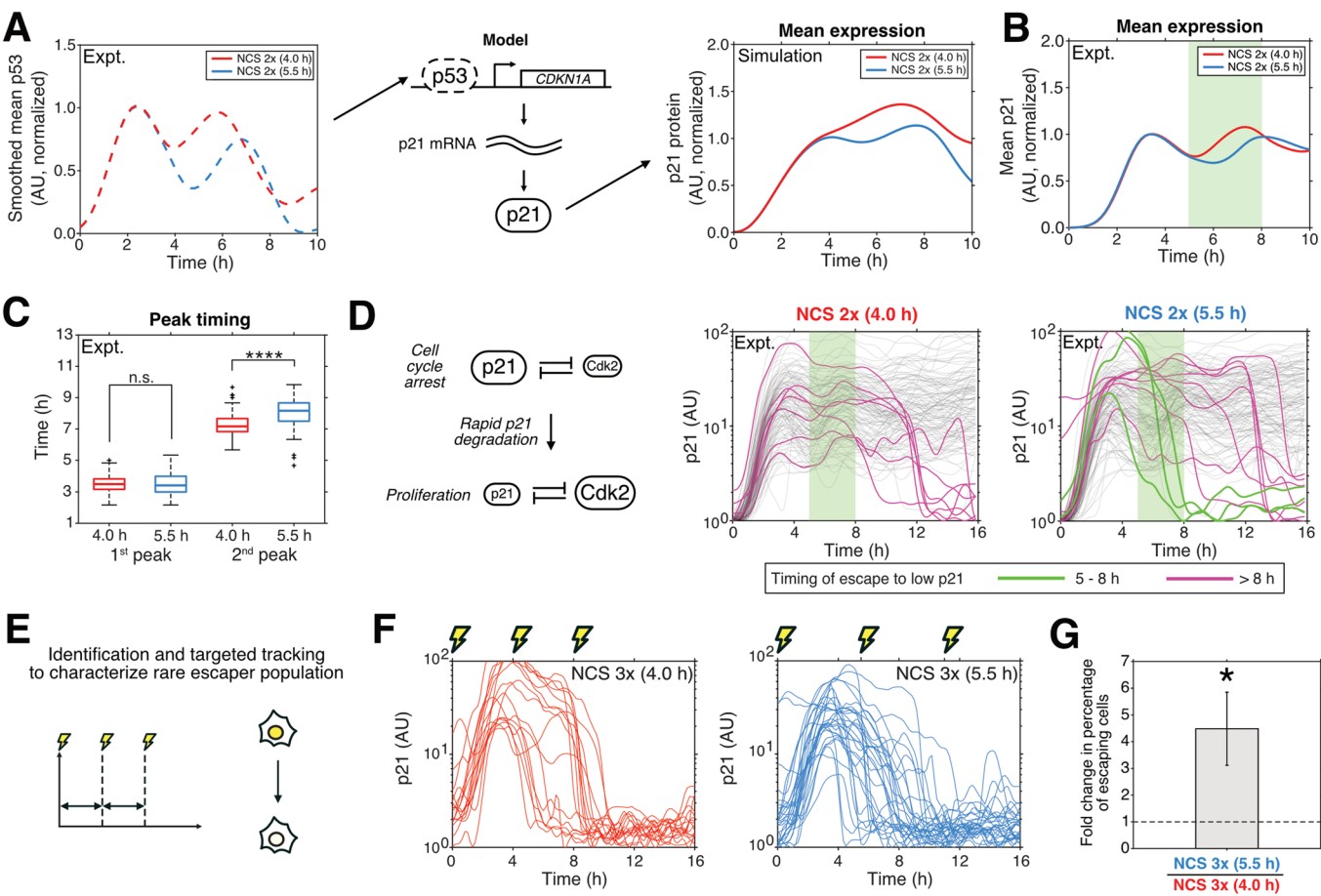

**Figure 4. Phase resetting of p53 pulses modulates downstream p21 expression and robustness of cell cycle arrest.**

(A) Mean p53-mVenus expression under two doses of NCS spaced 4.0 h (red) or 5.5 h (blue) apart, rescaled from 0 to 1. Traces shown were obtained by averaging the 4.0 h and 5.5 h conditions from 0 h to 4 h and smoothing for continuity (left). These traces were input into the mathematical model (middle) to predict downstream p21 expression (right) for the experimentally measured p53. (B) Experimentally measured mean p21 expression for the two conditions shown in (A). Experimentally observed curves are comparable to computationally predicted curves in (A, right). (C) Comparison of p21 peak timings for the first and second peaks in the 4.0 h and 5.5 h ($n = 135$ and 148, respectively; measured across three biological replicates) conditions showing no differences for the first peak, but a later second peak for the 5.5 h condition (****$P < 0.0001$; Wilcoxon rank-sum test). The center, upper and lower bounds, and upper and lower whiskers of the box plots indicate median, 75th and 25th percentiles, and the maximum and minimum data values, respectively. Crosses outside the whiskers represent outliers. (D) Schematic of cell cycle regulation by p21 and its toggle switch partner Cdk2 wherein rapid p21 degradation is indicative of cell cycle re-entry. Corresponding traces of p21 expression in NCS 4.0 h and NCS 5.5 h conditions demonstrating escape to a low p21 state either between 5.0 and 8.0 h (green) or after 8.0 h (magenta). Traces filtered for escapers prior to 5.0 h. Light green highlight represents the time period from 5.0 to 8.0 h. (E) Experimental workflow to better characterize rare escaper cells under different phase-resetting frequencies by increasing the active phase-resetting durations and specifically quantifying escapers. (F) p21 expression in single cells treated 3× with NCS at an interval of 4.0 h (left) or 5.5 h (right) showing the characteristic rise in expression upon DNA damage, followed by rapid degradation indicating cell cycle re-entry. Lighting marks indicate the timing of NCS treatment. Data is the cumulative of three biological replicates. (G) Fold change in the number of escaping cells in the 5.5 h interval treatment compared to the 4.0 h interval treatment. Bar graph and error bars represent the mean and standard deviation of three biological replicates. Symbol on top of the bar represents the statistical significance using a t test on the log fold changes with the null hypothesis of a log fold change of zero (*$P < 0.05$. Exact $P$ values provided in Table EV2).

widened synchronization range in the p53 system is due to the presence of a less rigid endogenous oscillator than predicted by the mathematical model (Fig. 3). While we focused here on perturbation of parameters related to the kinase ATM (Fig. 3), variation of multiple system parameters, including the MDM2 degradation rate constant, can generate similarly increased synchronization by reducing the rigidity of this oscillator. This rigidity-dependence of synchronizability is consistent with what has also been observed in the circadian system (Abraham et al, 2010; Bordyugov et al, 2011).

Energy landscape analysis of this phenomenon showed that a landscape which permitted more uniform cell behavior through

phase resetting was characterized by a wider stable steady-state valley under a constant input, a feature associated with more heterogeneity. In other words, a system that exhibits noisier oscillations under a constant input is expected to be better synchronized under periodic inputs. While this seems counterintuitive, it is consistent with the expected behavior of a weak attractor. Such an attractor permits significant noise-driven deviations from the mean under steady state (a key feature of p53 oscillations under DSBs (Reyes et al, 2018; Geva-Zatorsky et al, 2006)) while simultaneously being more malleable to an external cue, allowing for a more uniform response under external perturbations. Taken together, these observations suggest that

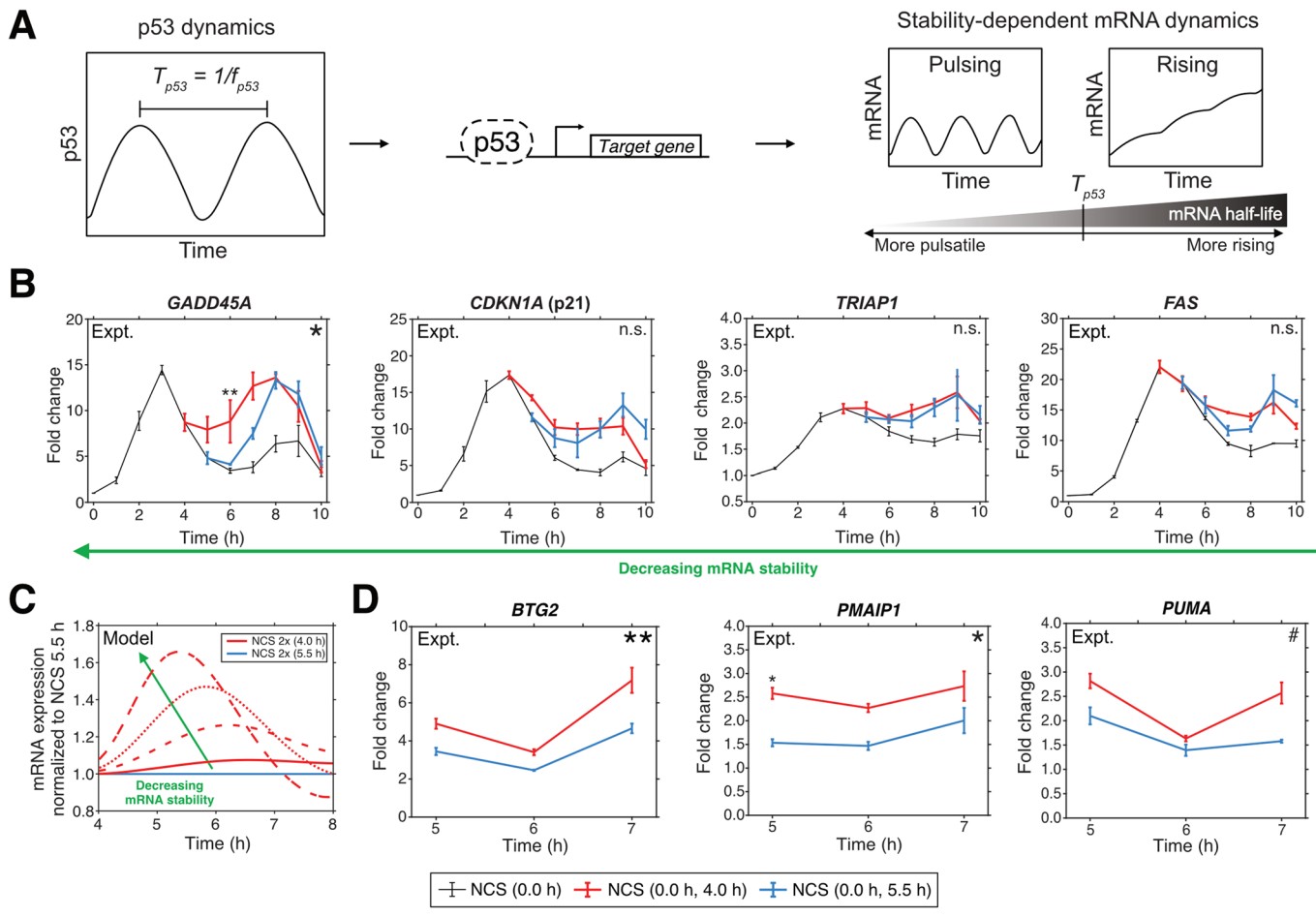

**Figure 5. Phase resetting of p53 pulses modulates downstream target expression in a degradation rate-dependent manner.**

(A) Schematic showing the effect of p53 pulse frequency and target mRNA degradation rate in encoding downstream gene expression patterns under p53 oscillations. Less stable targets follow the oscillations of p53 while more stable targets tend to accumulate with each pulse. (B) Gene expression of *GADD45A*, *CDKN1A*, *TRIAP1*, and *FAS* (mRNA decay rate constants of 0.599, 0.248, 0.212, and 0.050 h⁻¹, respectively) as measured by qRT-PCR in cells treated with a single dose (black lines), or double dose of NCS at 4.0 h or 5.5 h intervals (red or blue lines, respectively). (C) Mathematical modeling prediction of the difference in mRNA time integrals between 4.0 h and 8.0 h with an NCS double dose in a 4.0 h interval compared to a 5.5 h interval as a function of mRNA degradation rate constants. Rate constants from lowest to highest are $10^{-2.0}$, $10^{-1.5}$, $10^{-0.5}$, and $10^{0.0}$. (D) Gene expression of *BTG2*, *PMAIP1*, and *PUMA* (mRNA degradation rate constants of 0.844, 0.730, and 0.558 h⁻¹, respectively) in cells treated with a double dose of NCS at 4.0 h or 5.5 h (red and blue lines, respectively). For the plots in (B, D), the line plot represents the mean of three biological replicates and error bars indicate standard error of the mean ($^{#}P = 0.051$; $^{*}P < 0.05$; $^{**}P < 0.01$; Symbols at the top right indicate statistical significance of the effect of treatment interval as calculated by a repeated measures ANOVA and symbols above individual points indicate the statistical significances at each time point as obtained using a post-hoc multiple comparison test. Exact $P$ values provided in Table EV2).

p53 dynamics in response to DSBs are governed by a weak oscillator and that this feature is responsible for better synchronization of the system. However, it is unclear whether this can be simply recapitulated by reparameterization of the model or if it requires the addition of missing regulatory interactions. For example, incorporating bidirectional regulation of DSB repair and p53 could lead to better synchronization if there is a rigidity mismatch between these two interconnected oscillators (Gu et al, 2019).

Finally, we found that controlling p53 dynamics through phase resetting enabled modulation of the expression of downstream targets of p53, including targets involved in cell fate determination. When examining p21 expression in single cells (Fig. 4), we found that a higher driving frequency resulted in greater mean p21 expression across the population, which enabled better

maintenance of the cell cycle arrest-associated high-p21/low Cdk2 state compared with the effects of a lower driving frequency. However, the effects on other p53 targets were not consistent. Out of four initial downstream targets examined (Fig. 4E), only the least stable species *GADD45A* (involved in DNA repair) reproducibly altered expression dynamics at the population level. The effect of synchronizing treatments was qualitatively, but not statistically, reproducible in the more stable targets including *CDKN1A* (p21 mRNA), which exhibited stimulus frequency-dependent protein dynamics at the single-cell level. Mathematical modeling attributed this lack of statistical significance to a diminished difference in mRNA expression dynamics between the two input frequencies for more stable targets (Fig. 4F). These differences were likely more difficult to recapitulate experimentally due to natural biological variability, especially with bulk measurements of a non-

homogeneous population. This was subsequently validated by statistically significant observed increases in gene expression in three similarly unstable targets–BTG2 (cell cycle arrest), *PMAIP1* (pro-apoptotic), and *PUMA* (pro-apoptotic)—whose mRNA decay rate constants are much greater than the p53 pulse frequency (Porter et al, 2016). Overall, our findings imply that less stable p53 targets, such as those involved with DNA repair, apoptosis, and cell cycle arrest (Jiménez et al, 2022a; Porter et al, 2016; Hafner et al, 2017), are most likely to be affected by phase resetting. In general, these findings suggest externally driving p53 pulses at a frequency faster than those of a specific target transcript's decay rate can switch the target dynamics from a "pulse generator" (in which transcripts are periodically expressed and degraded) to an "integrator" of the stimulus (in which the transcript continues to accumulate in expression), all with the added effect of increasing synchronization of the downstream responses across cells. Our previous work has shown that specific target gene promoters can also exhibit additional regulatory mechanisms, including band-pass filtering and low-pass filtering of p53 oscillations (Harton et al, 2019), that would likely need to be taken into consideration when attempting to generate specific downstream cellular affects through phase resetting of p53 dynamics.

In comparison to other biological oscillators such as the circadian rhythm and NF-κB, the selective pressures leading to synchronized p53 oscillations are less obvious. It is possible that phase resetting is simply a consequence of the evolution of oscillations themselves in p53; this dynamic behavior has increased information transfer capacity in comparison to single-value steady states under noise and can therefore enable cells to respond sensitively to DNA damage while maintaining the potential to recover (Cheong and Levchenko, 2010; Reyes et al, 2018). Alternatively, given the effects of p53 on circadian rhythm regulation (Miki et al, 2013), metabolism (Humpton and Vousden, 2016), and NF-κB (Bohuslav et al, 2004; Murphy et al, 2011), it is possible that synchrony in p53 oscillations shapes more effective cellular responses in these contexts despite not being required for DNA repair. Examining the interplay of these oscillatory systems and p53 in regulating gene expression and cell fate determination could help elucidate the evolutionary advantages of phase resetting in p53 oscillations.

From a translational perspective, the increase in cell-to-cell synchrony in p53 dynamics observed here offers an avenue for designing cancer therapies that can result in more effective tumor cell clearance. Strategies involving phase resetting can mitigate effects such as fractional survival of tumors by providing a more homogenous initial population that is expected to respond more uniformly to a secondary therapy (Chen and Lahav, 2016). Further, the ability to modulate downstream gene expression through stimulus frequency can be used to bias cellular response towards specific cell fates—for example, stimulating p53 at frequencies that result in increased DNA repair gene expression and better maintenance of cell cycle arrest can aid in better recovery of cells in cases such as damage due to mustard gas exposure (Shalwitz et al, 2024).

Overall, this study demonstrates phase resetting of p53 oscillations in response to DSBs, elucidating fundamental oscillator characteristics that underpin this phenomenon. This method of synchronization also led to stimulus frequency-dependent modulation of downstream target genes responsible for cell fate

determination. Future work focused on utilizing the reduced single-cell heterogeneity as well as differential gene expression demonstrated here may aid in the design of more effective therapeutic strategies in contexts where p53 modulates cellular behavior such as cancer and chemical damage remediation.

# Methods

**Reagents and tools table**

| Reagent/resource | Reference or source | Identifier or catalog number |
|---|---|---|
| **Experimental models** | | |
| MCF-7 p53-mVenus | Batchelor et al, 2017 | |
| MCF-7 p53-mVenus H2B-Cerulean | This paper | |
| MCF-7 p21-mVenus DHB-mCherry H2B-Cerulean | Moser et al, 2018 | |
| HEK-293T | ATCC | CRL-3216 |
| **Recombinant DNA** | | |
| pLentiPGK-Hygro-DEST-H2B-mCerulean3 | Regot et al, 2014; Addgene | 90234 |
| **Antibodies** | | |
| **Oligonucleotides and other sequence-based reagents** | | **All primer sequences provided as 5'–3'** |
| BAX forward primer | Porter et al, 2016 | CTGACGGCAACTTCAACTGG |
| BAX reverse primer | Porter et al, 2016 | GATCAGTTCCGGCACCTTGG |
| BBC3/PUMA forward primer | Porter et al, 2016 | CGACCTCAACGCACAGTACG |
| BBC3/PUMA reverse primer | Porter et al, 2016 | GGGTGCAGGCACCTAATTGG |
| BTG2 forward primer | Porter et al, 2016 | AGGCACTCACAGAGCACTAC |
| BTG2 reverse primer | Porter et al, 2016 | TGGGGTCCATCTTGTGGTTG |
| CDKN1A forward primer | Porter et al, 2016 | TACCCTTGTGCCTCGCTCAG |
| CDKN1A reverse primer | Porter et al, 2016 | ATCAGCCGGCGTTTGGAGTG |
| FAS forward primer | Porter et al, 2016 | CCCGGACCCAGAATACCAAG |
| FAS reverse primer | Porter et al, 2016 | TGTTCACATTTGGTGCAAGGG |
| GADD45A forward primer | Porter et al, 2016 | TGCTGGTGACGAATCCACATT |
| GADD45A reverse primer | Porter et al, 2016 | TGATCCATGTAGCGACTTTCCC |
| GAPDH forward primer | Porter et al, 2016 | ACATCGCTCAGACACCATG |
| GAPDH reverse primer | Porter et al, 2016 | TGTAGTTGAGGTCAATGAAGGG |

| Reagent/resource | Reference or source | Identifier or catalog number |
|---|---|---|
| PMAIP1/NOXA forward primer | Porter et al, 2016 | GCAAGAACGCTCAACCGAG |
| PMAIP1/NOXA reverse primer | Porter et al, 2016 | TCCTGAGCAGAAGAGTTTGGAT |
| TRIAP1 forward primer | Porter et al, 2016 | GACATGAAGCGCGAGTACGA |
| TRIAP1 reverse primer | Porter et al, 2016 | ATTGCTTTCTGAACACACTGCT |
| **Chemicals, enzymes, and other reagents** | | |
| Neocarzinostatin | MilliporeSigma | N9162 |
| RNeasy Plus Mini Kit | QIAGEN | 74134 |
| QIAShredder | QIAGEN | 79654 |
| High-Capacity cDNA Reverse Transcription Kit | ThermoFisher | 4374966 |
| iTaq Universal SYBR Green Supermix | Biorad | 1725121 |
| RPMI media | Fisher Scientific | SH3002701 |
| RPMI with no phenol red | ThermoFisher | 11835030 |
| Lenti-X Packaging Single Shot (VSV-G) | Takara | 631275 |
| **Software** | | |
| MATLAB | Mathworks | https://www.mathworks.com/products/matlab.html; RRID:SCR_001622 |
| ImageJ | NIH | https://imagej.nih.gov/ij/; RRID:SCR_003070 |
| NIS Elements | Nikon | https://www.microscope.healthcare.nikon.com/products/software/nis-elements, RRID:SCR_014329 |
| p53Cinema | Reyes et al, 2018 | |
| **Other** | | |
| 96-well Glass Bottom No. 1.5 plate | Mattek | P96G-1.5-5-F |

## Mathematical modeling of p53 dynamics

Deterministic simulations were carried out in MATLAB (Mathworks, Natick, MA, USA), and stochastic simulations were carried out using the Gillespie algorithm (Gillespie, 1976) in C++. A mathematical model of the p53 response in response to double-stranded DNA breaks (DSBs) developed by Mönke et al (Mönke et al, 2017) was used in all simulations, with modifications as specified. The mRNA synthesis rates of MDM2 and WIP1 were increased by 20% each (from 1.0 AU/h to 1.2 AU/h) to increase the observed p53 pulse frequency from once every 7.0 h to once every 5.5 h, matching the expected frequency in MCF-7 cells (Geva-Zatorsky et al, 2006). Other parameters could also be potentially varied to get this same effect. We chose MDM2 and WIP1 production rates based on the sensitivity analysis by Mönke et al

(Mönke et al, 2017). Additionally, the functional form of DSB sensing by ATM, $f(DSB)$ (Fig. EV6A), was changed from a saturable equation, $f(DSB) = DSB/(\gamma + DSB)$, to a non-saturable one, $f(DSB) = log(DSB/\gamma + 1)$, to reflect the shift from oscillatory to sustained p53 dynamics observed experimentally under high DNA damage (Batchelor et al, 2008). Both of these functions show very similar behavior for small numbers of DSBs (Fig. EV6B). However, the saturable function approaches a maximum of 1 and continues to drive oscillatory behavior for large numbers of DSBs (Fig. EV6C). In contrast, the non-saturable log function causes a shift from oscillatory to sustained p53 dynamics under high damage (Fig. EV6D). Values for all rate constants used are provided in Table EV1.

### Deterministic model

The double-stranded break and repair process from the original model (Mönke et al, 2017) was changed from a discrete stochastic birth-death process to a deterministic equation: $d(DSB)/dt = b - r*DSB$. Following Mönke et al (Mönke et al, 2017), for the first hour after NCS treatment, the birth rate was changed to $b_s$, a parameter proportional to the NCS concentration. After one hour until further damage was inflicted, the birth rate reverted to a basal birth rate $b_b = 2.3$ breaks/h. In addition, DSBs were repaired with a constant death rate $r = 0.315\,h^{-1}$ throughout the simulation. For repeated treatment, this resulted in a sawtooth-like waveform where there was nearly linear accumulation of DSBs with a slope proportional to $b_s$ for 1 h followed by an exponential decay (Fig. EV6E). Simulation files are provided in the online data repository (https://doi.org/10.17632/xrc7k83tjv.1).

### Stochastic model

The model included two key sources of noise: (1) DSB birth-death process, and (2) mRNA production and degradation. Due to its independence from the rest of the model, noise due to stochastic DSB generation and repair was implemented by separately generating the appropriate waveform using the Gillespie method (Gillespie, 1976), and then feeding this waveform as a piecewise function of time into the time-stepping algorithm. The mean of the DSB process realizations closely followed the sawtooth-like deterministic shape (Fig. EV6F). The model with DSB noise was simulated in MATLAB R2020a using the *ode23s* function with the DSB waveform input as a piecewise function such that for $t_i < t < t_{i+1}$, $DSB = DSB_i$, where $DSB_i$ is the number of double-strand breaks as per the stochastic birth-death model. To include noise from the stochastic production and degradation of mRNA species (WIP1 and MDM2), we implemented a hybrid stochastic-deterministic simulation in C++. In this implementation, the equations for WIP1 and MDM2 mRNAs were solved using the Gillespie method and all other equations were solved deterministically using discrete time steps between each time step of the Gillespie algorithm. The number of time steps was chosen to have a smaller average time step duration than the DSB noise-only simulation from MATLAB. The number of DSBs was updated according to the pre-generated waveform such that $DSB = DSB_i$ when $t > t_i$, $DSB = DSB_{i+1}$ when $t > t_{i+1}$, and so on. Each stochastic simulation was run 1000 times, as determined by the convergence in synchronization score (Fig. EV6G). Simulation files

are provided in the online data repository (https://doi.org/10.17632/xrc7k83tjv.1).

## Characterization of synchronization

For deterministic simulations, the system was considered synchronized to the input if, at steady state, the period was within 0.1% of the input period (an order of magnitude lower than the period grid size/interval when evaluating synchronization). Due to variability in the period of oscillation in stochastic systems, this approach was not applied there. Instead, the degree of synchrony was calculated using the synchronization score ($S$), following the procedure used by Gupta et al (Gupta et al, 2016). Briefly, this score is the fraction of power at the input frequency and its harmonics, compared to the entire frequency spectrum as calculated by a one-sided discrete Fourier transform (DFT, custom script in MATLAB). The DFT calculates power $P$ at a set of frequencies $f$ dependent on trajectory length and discretization. Due to finite observation time, we will have a finite number of frequencies which may or may not include the exact input frequency and the corresponding harmonics. Instead, the spectral power of the system will be spread around the components of the DFT that are near the exact frequencies. In consideration of this numerical issue, we used a frequency window $w$, that was 1% of the width of the natural frequency of the system $f_{natural}$ within which the corresponding power for each frequency was calculated. The synchronization score was then calculated as follows,

$$S = P_{in}/P_{total} \tag{1}$$

where,

$$P_{in} = \sum_i \sum_k P(f_k : (i * f_{in} - w) \leq f_k \leq (i * f_{in} + w)) \tag{2}$$

$$P_{total} = \sum_k P(f_k) \tag{3}$$

$$w = 0.01 * f_{natural} \tag{4}$$

$f_k$ is the $k$th frequency of the DFT, $i$ is an integer corresponding to the $i$th harmonic, $f_{in}$ is the input frequency, $w$ is the window size around the frequencies of interest, and $f_{natural}$ is the natural frequency of the system.

All simulations were run for 5000 h for characterization. As opposed to shorter times, this duration exhibited minimal fluctuation in the synchronization score in phase-locked regions (Fig. EV6H). These fluctuations are artifacts caused by the finite numerical nature of the DFT, as explained in the previous paragraph.

## Parameter analysis for system synchronization

### Overview

Parameters were varied, one at a time, and the resulting system was classified as oscillatory or non-oscillatory based on the total Fourier power of the normalized waveform. The system was then rescaled by multiplication with 5.5 (natural period of the basal system) and division with the natural period of the modified system.

Synchronization, as defined by phase-locking, was examined in a window of ± 2 h around the natural pulsing period of the system.

### Parameter variation

Simulations of the fully deterministic system were used to characterize the effect of individual parameters on synchronization. Each parameter was separately varied in a tenfold range below and above the default value in increments of 0.005 on the $\log_{10}$ scale.

### Constant DSB simulation and classification into oscillatory/non-oscillatory

The system was first simulated for a constant DSB input of 200 for $t = 5000$ h. The p53 trajectory after 1000 h (in intervals of 0.05) was classified as oscillatory or non-oscillatory based on the total power on the Fourier spectrum. The trajectory was first rescaled linearly from 0 to 1 (*rescale* function in MATLAB) to remove variability in power due to amplitude and then was used to calculate the one-sided DFT. If the total power of the DFT was >0.2, it was classified as oscillatory. This threshold was determined by simulating 1000 Latin-hypercube sampled parameters in a twofold range, for constant DSB inputs from 0 to 1000 in increments of 20, and examining the distribution of total power (Fig. EV6I). A clearly bimodal distribution arose: one population of trajectories had a minimum power of 0.2, likely representing oscillatory cases, while the other population had a maximum power of ~0.1, corresponding to non-oscillatory cases. A few randomly selected trajectories from both parts of the distribution were visually inspected to validate this. Non-oscillatory cases were discarded while oscillatory cases moved on to the next stage.

### System rescaling with natural period

Oscillatory systems were then rescaled by multiplying the right-hand side of the differential equation expressions by 5.5 h (natural period of the basal system) and dividing it by the natural period of the oscillatory system (in h). This ensures commensurate comparisons of synchronization ranges across different parameter sets where the system may have different natural periods.

### Synchronization curve generation and characterization

The natural period of the oscillatory trajectory was determined by the most prominent peak on the DFT. A window (± 2 h, in increments of 0.05 h) was defined around this period and synchrony was characterized for periodic inputs by examining phase-locking within this region. The effect of each parameter on synchrony was examined by the number of points with phase-locking for each parameter value divided by the total number of points in the window.

## Mathematical modeling of p21 dynamics

p21 protein dynamics in Fig. 4B were simulated using a mathematical model of p21 mRNA and protein dynamics developed by Reyes et al (Reyes et al, 2018). All rate constants were the same as the original publication except for $K_0$ (Michaelis constant of p53-dependent p21 transcription) which was set to 0.67. This was done to account for the differences in the scaling of p53 traces between this work and Reyes et al (Reyes et al, 2018). Simulation files are provided in the online data repository (https://doi.org/10.17632/xrc7k83tjv.1).

## Mathematical modeling of downstream targets

General mRNA dynamics in Fig. 4H were obtained using a standard p53-dependent transcription model with p53 levels from Fig. 4A rescaled from 1 to 5 as the input, consistent with Porter et al (Porter et al, 2016). Rate constant values were the same as in the original work.

$$\frac{d(mRNA)}{dt} = p53 - k_d * mRNA \qquad (5)$$

Simulation files are provided in the online data repository (https://doi.org/10.17632/xrc7k83tjv.1).

## Cell culture

MCF-7 cells transfected with p53-mVenus and H2B-Cerulean were maintained in RPMI-1640 media supplemented with 10% (v/v) FBS and 1% (v/v) Pen-Strep (all reagents from Life Technologies) in a humidified incubator at 37 °C and 5% $CO_2$. MCF-7 cells with p21 tagged fluorescently at the endogenous genomic locus (a kind gift from Sabrina L Spencer (Moser et al, 2018)) were maintained in the same medium. Due to the loss of DHB-mCherry (Cdk2 sensor) and/or H2B-mTurquoise (nuclear marker) in subpopulations of these cells, fluorescence-activated cell sorting (FACS) using a BD FACSAria II instrument was used to obtain cells that retained expression of both constructs. Cell lines were low passage number and validated as exhibiting the expected fluorescent reporter activity.

## Fluorescent clonal cell line generation for live-cell imaging of p53 expression

Lentiviral particles containing pLentiPGK-Hygro-DEST-H2B-mCerulean3 vector (kind gift from Markus Covert (Regot et al, 2014); Addgene: 90234) were produced in HEK-293T (ATCC) cells using Lenti-X Packaging Single Shots (TakaraBio) according to the manufacturer's protocol. The resulting particles were used to infect MCF-7 cells expressing a previously characterized p53-mVenus fusion (Batchelor et al, 2008). Cells stably expressing both constructs were selected in cell culture media containing 50 µg/mL hygromycin and 400 µg/mL neomycin (Life Technologies). Clonal cell lines were generated by limiting dilution. Clones were then screened for the presence of Cerulean fluorescence. In addition, the cells were validated to exhibit the previously reported pulsatile p53-mVenus expression dynamics under 400 ng/mL of neocarzinostatin (NCS; MilliporeSigma) and routinely tested for mycoplasma contamination.

## Live-cell imaging and single-cell tracking

For live-cell imaging, cells were plated on 96-well No. 1.5 glass-bottom dishes (Mattek) to be 60% confluent at the start of treatment. Prior to imaging, cells were washed with DPBS (1×), after which transparent RPMI-1640 media (Life Technologies)—lacking phenol red, riboflavin, and L-glutamine, and supplemented with 2% (v/v) FBS and 1% (v/v) Pen-Strep – was added. Cells were imaged with a Nikon Eclipse TiE inverted fluorescence microscope equipped with an automated stage (Prior) and a custom chamber to maintain 37 °C, 5% $CO_2$, and high humidity. Images were collected every 10 min for the brightfield, YFP, and CFP channels using a 20x CFI Plan Apochromat Lambda (NA = 0.75) objective (Nikon). Exposure time for each channel was set to be <500 ms. The ND2 format imaging data were exported as individual tiff files for each channel, time point, and stage position using Bio-Formats (Linkert et al, 2010) command line tools. Cells were tracked semi-automatically in p53Cinema (Reyes et al, 2018) using the nuclear marker H2B-mCerulean in the CFP channel, agnostic to p53-mVenus expression in the YFP channel. p53-mVenus expression for each cell track was then obtained using the getDatasetTraces_fillLineageInformation function with a sampling radius of 5 pixels applied to the YFP channel. The same procedure was followed to obtain p21-mVenus expression in the 2× repeated dose analyses of p21-tagged MCF-7 cells. In case of the extended 3× repeated dose analyses, cells that had visible transient upregulation of p21-mVenus expression within the 12-h analysis period were identified manually and then specifically tracked using p53Cinema. Nuclear p21-mVenus and DHB-mCherry expression were obtained using the getDatasetTraces_fillLineageInformation function with a sampling radius of 5 pixels applied to the YFP and TxRed channels, respectively. Total responding cell counts at each position for this analysis were obtained using ImageJ by applying a peak-detection algorithm on Gaussian-blurred p21-mVenus images at $t = 3.0$ h. The thresholding for the peak detection was set such that the responding cell counts approximately matched the number of responding cells counted manually for a selected set of images.

## p53 single-cell trace processing and analysis

All p53-mVenus traces for each stage position were first divided by the mean p53-mVenus fluorescence reading at the start of the experiment ($t = 0$) at that position to account for illumination variability between stage positions. Any missing readings were replaced by the p53-mVenus expression at the previous time point within that trace (*fillmissing* function in MATLAB). The traces were then smoothed using a gaussian moving filter (*smoothdata* function in MATLAB) with a window size corresponding to a real-world time of 3 h and 20 min; this window size aids in smoothing out smaller noisy fluctuations while retaining overall pulsatility of the system. The final traces were obtained by subtracting the minimum value of each trace from itself providing a lower bound at 0 for the p53 signal in each cell. The analysis script is provided in the online data repository (https://doi.org/10.17632/xrc7k83tjv.1).

## Peak detection and peak timing analysis

Peaks were detected for each trace and the mean of all traces using the *findpeaks* function in MATLAB and specifying a set number of peaks to be detected. To quantify the peak timing variability for each *n*th peak, we first identified the peak in each trace that is closest to the *n*th peak of the mean p53 signal and then calculate the median absolute deviation in peak timing in this set. Using this method instead of simply considering the *n*th peak for each trace avoids erroneous propagation of confounding effects due to cells that did not pulse during the first dose, but rather pulsed starting from subsequent doses, while still accounting for whether a cell pulsed or not. The analysis script is provided in the online data repository (https://doi.org/10.17632/xrc7k83tjv.1).

## Gene expression quantification

Cells were plated on 35 mm dishes to be 60% confluent at the start of treatment. To obtain cell pellets, each dish was harvested at the specified time point by scraping in DPBS followed by centrifugation and aspiration of the supernatant. The resulting pellets were immediately stored at $-80\,^{\circ}$C. Total RNA was extracted from cell pellets using a RNeasy Mini Kit (QIAGEN). Lysis was performed using 350 μL of the provided lysis buffer, and the lysate was passed through a QIAshredder column (QIAGEN). RNA from the lysate was obtained by following the manufacturer's protocol and quantified using a Nanodrop 2000 (ThermoFisher Scientific). cDNA was prepared by loading 2 μg of RNA per sample into a 20 μL reverse transcription (RT) reaction using a High-Capacity cDNA Reverse Transcription Kit (ThermoFisher Scientific) as per the manufacturer's protocol. The resulting cDNA mixture was diluted 1:5 (20 μL of cDNA added to 80 μL of nuclease-free water) before usage. A 20 μL reaction mix was prepared using 10 μL iTaq Universal SYBR Green Supermix (Bio-Rad), 1 μL 10 μM forward/ reverse primer mix (sequences from Porter et al (Porter et al, 2016)), 8 μL nuclease-free water, and 1 μL template. qRT-PCR was then performed using a CFX Connect Real-Time PCR Detection System (Bio-Rad) with the following conditions: hot start (95 °C for 2 min), and 40 cycles of PCR (96 °C for 5 s, 64 °C for 20 s), ending in melt curve acquisition (64 °C–95 °C with 0.5 °C resolution). Melt curves (Change in relative fluorescence units per unit temperature vs. Temperature) were visually inspected for existence of a single peak to validate specific amplification for each gene. The threshold cycle count values ($Cq$) for each gene were obtained using the Bio-Rad CFX Manager 3.1 software. The fold change in expression was calculated by $2^{-\Delta\Delta C_q(X,t)}$ where $\Delta\Delta C_q(X,t)$ is defined as:

$$\Delta\Delta C_q(X,t) = \left[\overline{C}_q(X,t) - \overline{C}_q(GAPDH,t)\right] \\ - \left[\overline{C}_q(X,0) - \overline{C}_q(GAPDH,0)\right] \quad (5)$$

where $\overline{C}_q(X,t)$ is the mean $C_q$ value for gene $X$ at time $t$.

## Statistical analysis

All statistical analyses were carried out using GraphPad Prism 9.0 or MATLAB. In experimental design no explicit blinding was performed, although all single-cell analysis was performed using an automated pipeline to minimize sample bias and improve reproducibility. Experimental data in Figs. 1 and 2 passed a normality check using the Shapiro–Wilk test (Fig. EV6J) and were compared using one-way ANOVA followed by multiple comparison testing (Prism). Fold changes in variability for NCS 4.0 h ($n = 6$, originally) and 4.5 h ($n = 7$, originally) were run through an outlier detection test (Prism), and one and two data point(s) for each condition, respectively, were identified as outliers and discarded prior to further statistical analysis. Peak timings in Fig. 4E failed a normality check by using the Kolmogorov–Smirnov test and were therefore compared using a nonparametric Wilcoxon rank-sum test (*kstest* and *ranksum* functions in MATLAB). For qRT-PCR analysis, the fold change of each gene from three biological replicates was analyzed on the log scale using a repeated measures ANOVA, followed by multiple comparisons. A list of

exact p values for all statistical tests performed is provided in Table EV2.

## Data availability

The data sets and computer code produced in this study are available in the following database: Modeling computer scripts, imaging data sets, gene expression data sets, and data analysis scripts: Mendeley Data (https://doi.org/10.17632/xrc7k83tjv.1).

The source data of this paper are collected in the following database record: biostudies:S-SCDT-10_1038-S44320-025-00091-8.

## Peer review information

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

## Acknowledgements

The authors thank Kala Guettler for assistance with single-cell tracking. The authors also thank Sabrina Spencer for providing the p21-Cdk2 tagged cell line used in this study. This work was supported by funding from the University of Minnesota (SMA and EB) and the National Institutes of Health (R35GM136309 to CAS and R01GM149666 to EB), as well as access to high-performance computing resources from the Minnesota Supercomputing Institute and fluorescence-activated cell sorting services from the University Flow Cytometry Resource at the University of Minnesota.

## Author contributions

**Harish Venkatachalapathy**: Conceptualization; Resources; Software; Formal analysis; Investigation; Methodology; Writing—original draft; Writing—review and editing. **Samuel Dallon**: Investigation. **Zhilin Yang**: Resources; Methodology. **Samira M Azarin**: Conceptualization; Resources; Formal analysis; Supervision; Funding acquisition; Methodology; Writing—original draft; Writing—review and editing. **Casim A Sarkar**: Conceptualization; Resources; Formal analysis; Supervision; Funding acquisition; Methodology; Writing—original draft; Writing—review and editing. **Eric Batchelor**: Conceptualization; Resources; Formal analysis; Supervision; Funding acquisition; Methodology; Writing—original draft; Writing—review and editing.

Source data underlying figure panels in this paper may have individual authorship assigned. Where available, figure panel/source data authorship is listed in the following database record: biostudies:S-SCDT-10_1038-S44320-025-00091-8.

## Disclosure and competing interests statement

The authors declare no competing interests.

# Expanded View Figures

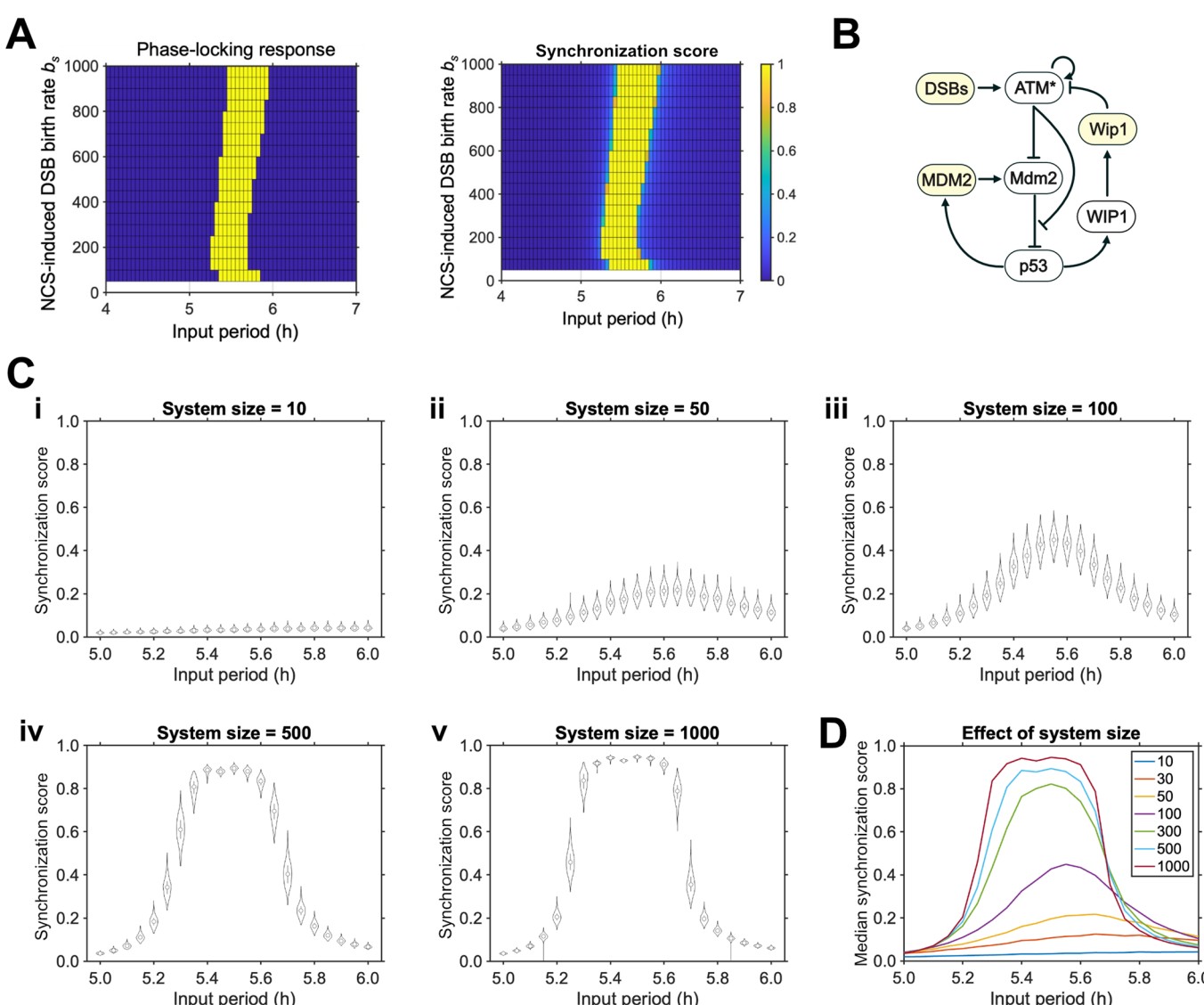

**Figure EV1.  Synchronization score characterization for the deterministic and stochastic p53 models.**

(A) Phase-locking response and synchronization scores as a function of NCS-induced birth rate and input period. Yellow regions represent phase-locking while blue represents no phase-locking. (B) Diagram of the p53 regulatory network responsive to DSBs. Yellow boxes denote species with stochasticity. (C) Violin plots of the top 99% of synchronization scores for $n = 1000$ stochastic simulations of the p53 DSB response in different system sizes at different NCS input periods for an NCS-induced break rate of 200 breaks/h. Here, system size refers to the factor used to convert concentration to number of molecules in the stochastic simulations. Larger system size implies more molecules and, therefore, less noise. Conversely, smaller system size implies fewer molecules and more noise. (D) Median synchronization score of 1000 stochastic simulations of the p53 DSB response in different system sizes at different NCS input periods for an NCS-induced break rate of 200 breaks/h. Overall, there is no increase in synchronization range due to intrinsic noise in comparison to the deterministic system.

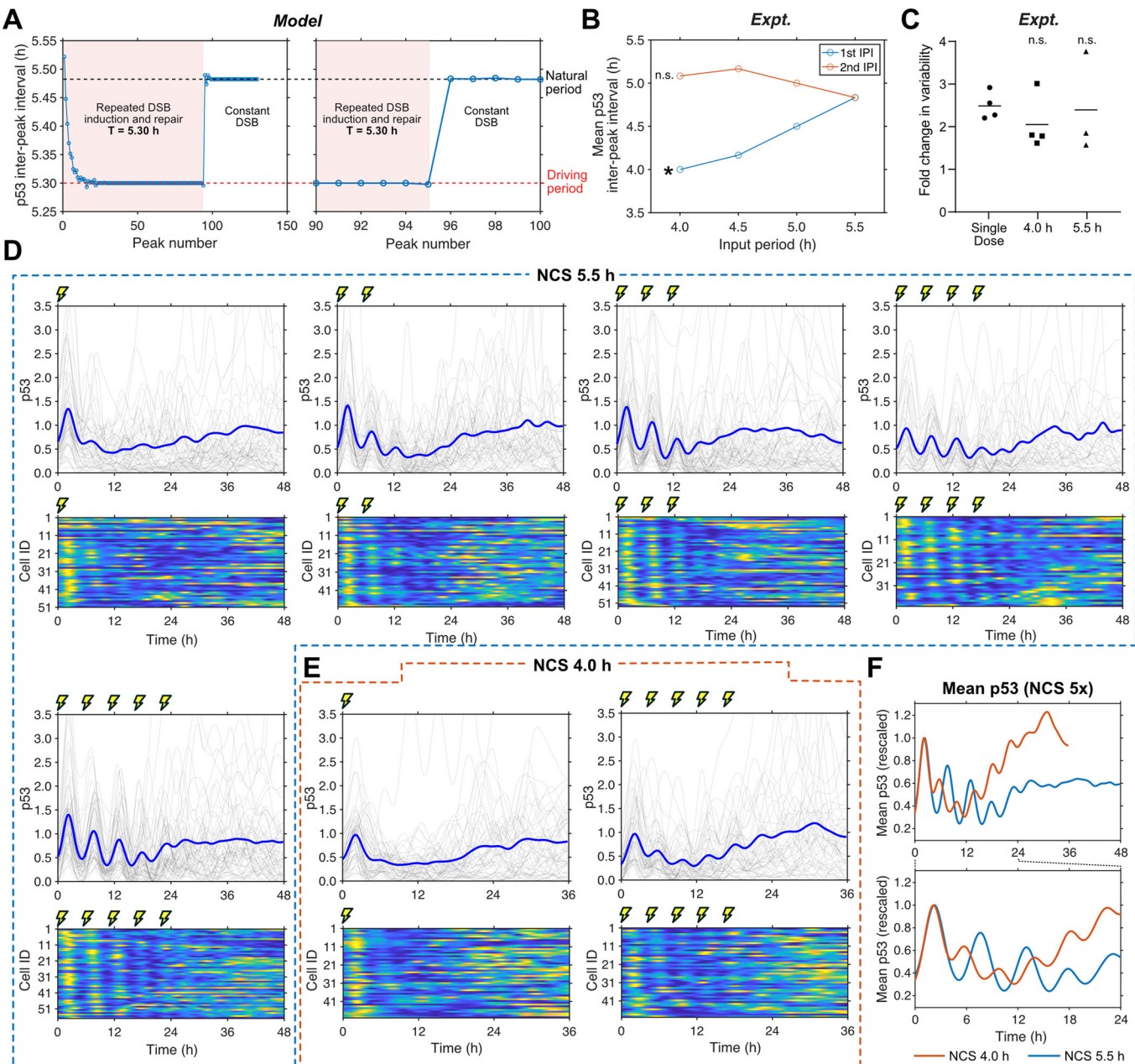

**Figure EV2.   p53 synchronization is consistent with phase resetting rather than entrainment.**

(**A**) Graph of the p53 interpeak interval in response to repeated DSB induction at a period of 5.3 h (red area) and after a shift to constant levels of DSB (white area). (**B**) Mean p53 interpeak interval as a function of the input period quantified from the data in Fig. 2 (*$P < 0.05$; $P$ value indicates the statistical significance of a non-zero slope in a linear regression vs. input period. Exact $P$ values provided in Table EV2). (**C**) Dot plots showing the fold change in peak timing variability when there is no phase-resetting dose. These are the 2nd to 1st peak fold changes for the single-dose case and the 3rd to 2nd peak fold changes for the 4.0 h and 5.5 h cases. ($n = 4$, 4, and 3 biological replicates from left to right; statistical significance calculated using an ANOVA). (**D, E**) Single-cell traces (gray) and mean (dark blue) line plots and the corresponding per-cell rescaled heatmaps of p53-mVenus expression under one to five doses of NCS (400 ng/mL) spaced 5.5 h apart (**D**, enclosed in the blue dashed outline) along with those from cells treated with one and five doses of NCS spaced 4.0 h apart (**E**, enclosed in the orange dashed outline). (**F**) Mean p53 expression of cells treated with five doses of NCS (400 ng/mL) spaced 4.0 h (orange) or 5.0 h (blue) apart showcasing the difference in pulse frequencies during the dosing regimens.

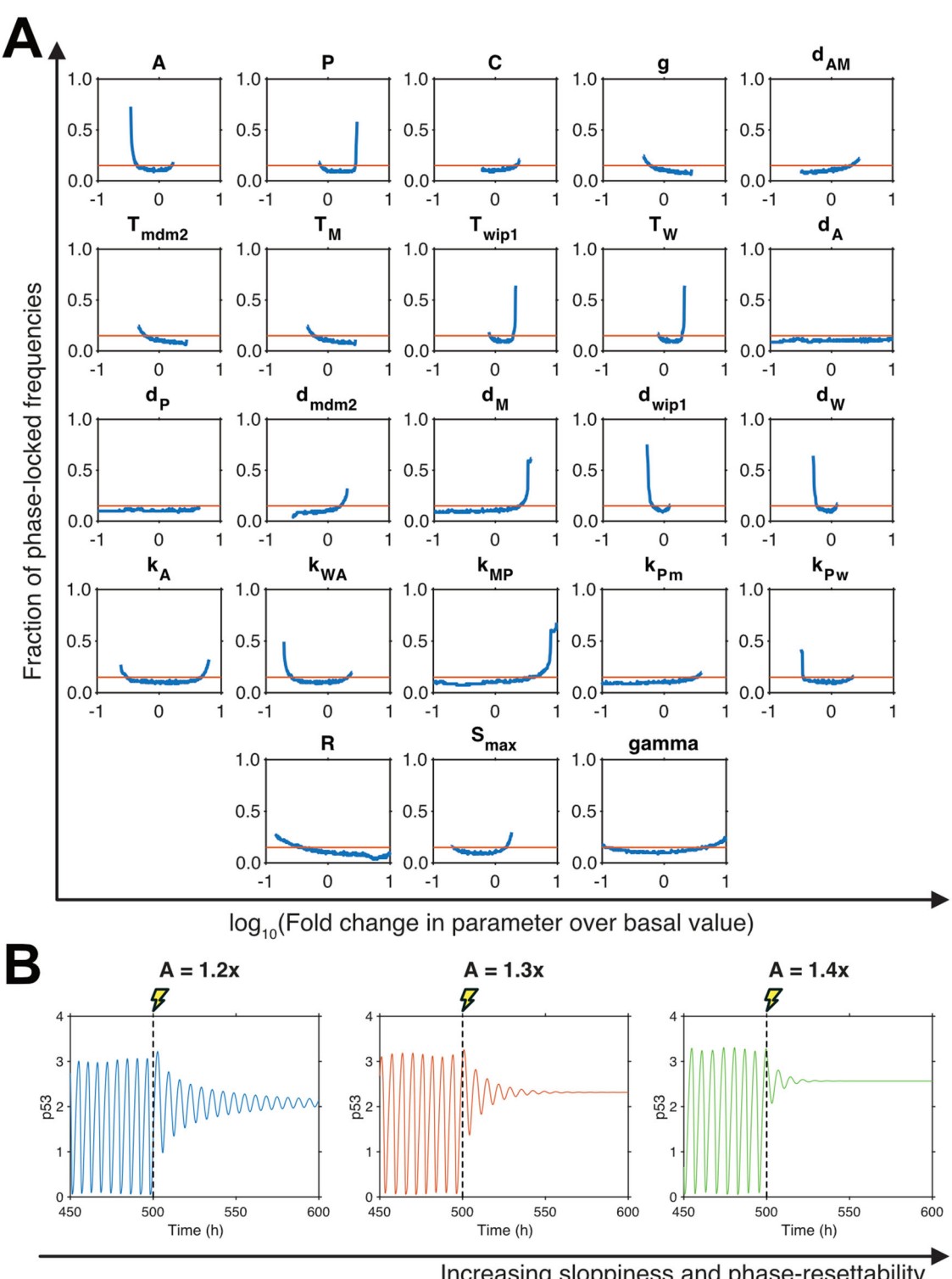

**Figure EV3. Effect of system parameters on synchronizability and transition to non-oscillatory regimes.**

(A) Effect of $\log_{10}$ fold change in each parameter on the fraction of input periods in a scaled window around the natural oscillatory period of the system. Effect of parameters considered significant if the y-axis value exceeded 0.15 (orange line, ~50% increase over basal synchronization range). Explanation of symbols provided in Table EV1. (B) Change in p53 dynamics due to a sudden and large sustained increase in levels of DNA damage at $t = 500$ h for different values of the ATM autophosphorylation rate constant (parameter A). A higher value of A results in a more phase-resettable system which corresponds to a faster decay in the transient oscillations in reaching the new stable steady state.

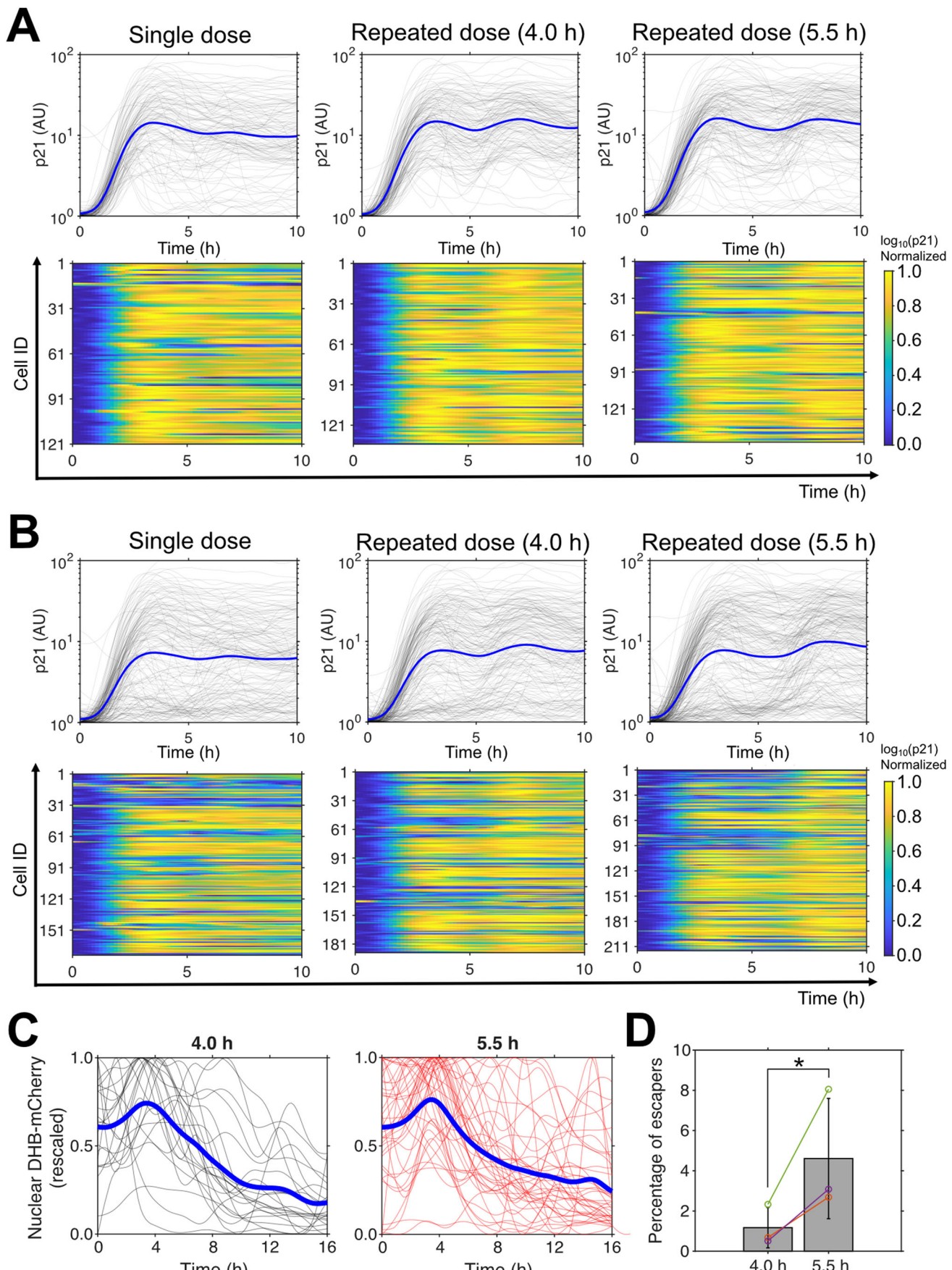

◀ **Figure EV4.  p21 expression in single cells treated with different NCS dosing regimens and Cdk2 activity in escaper cells.**

(A) Single-cell traces (gray) and mean p21 expression values (blue) for conditions from Fig. 4 along with a single dose of NCS as well as heat map diagrams for these conditions where each p21 trace is rescaled from 0 to 1. $n = 121$, 135, and 148 for the single dose, double dose (4.0 h) and double dose (5.5 h) conditions, respectively. (B) All single-cell traces of p21 expression without filtering for responding cells showing variability in timing of p21 induction despite the same treatment conditions (as reported previously (Sheng et al, 2019)). $n = 175$, 190, and 216 for the single dose, double dose (4.0 h) and double dose (5.5 h) conditions, respectively. (C) Nuclear DHB-mCherry fluorescence in escaper cells (rescaled from 0 to 1 on a per-cell basis) for the 4.0 h and 5.5 h 3× repeated NCS treatment regimes. (D) Percentage of escaper cells for the 4.0 h and 5.5 h 3× repeated NCS treatment regimes. Bar plots and error bars indicate the mean and standard deviation of three biological replicates. Each line plot corresponds to the percentage of escaper cells from one replicate. Statistical significance calculated by a $t$ test on the log fold change with a null hypothesis of log fold change $= 0$ (*$P < 0.05$. Exact $P$ values provided in Table EV2).

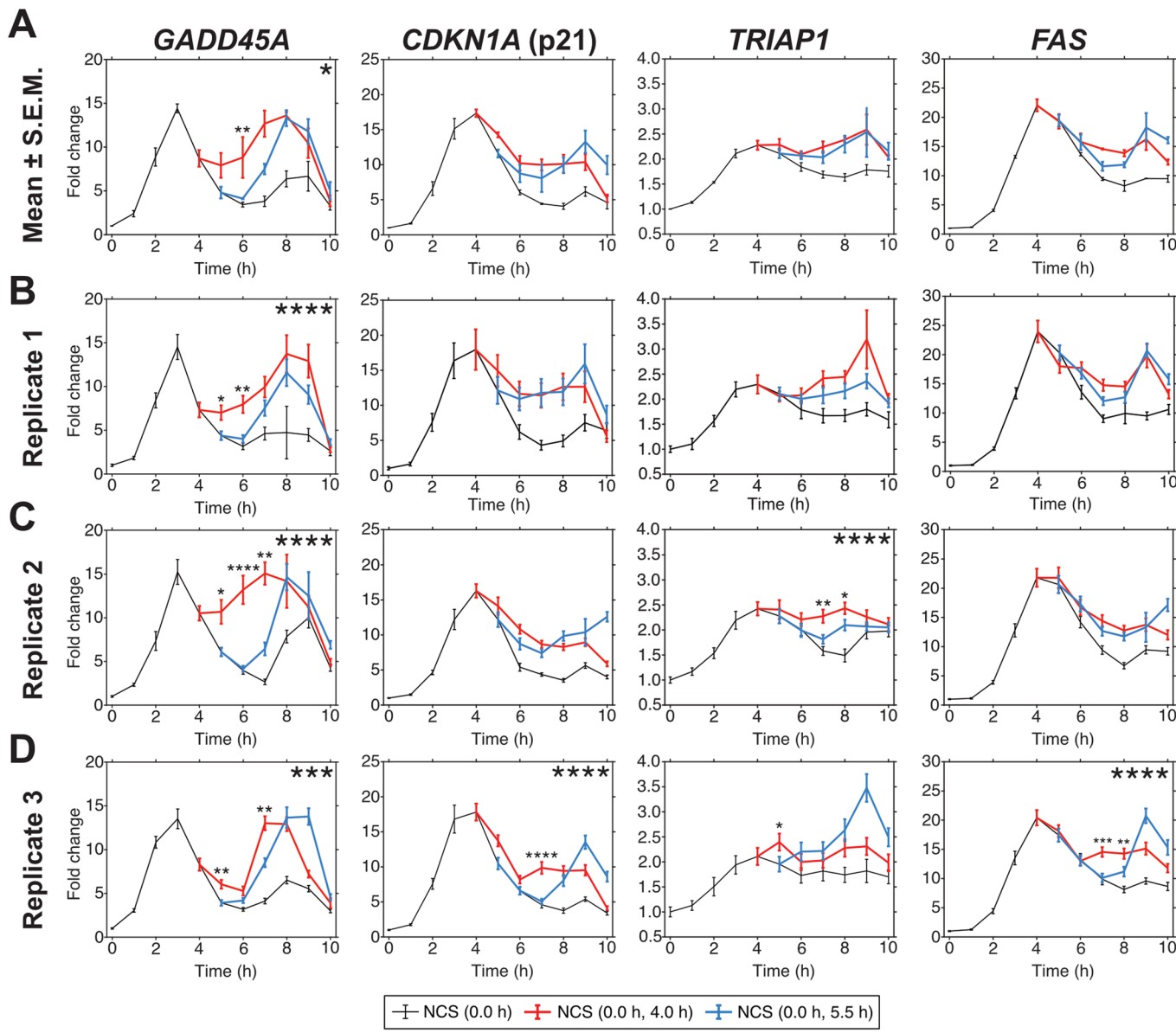

**Figure EV5.  p53 target gene expression as a function of NCS dose interval.**

Gene expression of *GADD45A*, *CDKN1A*, *TRIAP1*, and *FAS* as measured by qRT-PCR in MCF-7 p53-mVenus cells treated with a single dose (black lines) or double dose of NCS at 4.0 h or 5.5 h intervals (red or blue lines, respectively). (**A**) Mean and SEM of three biological replicates. (**B–D**) Mean and standard deviation of the three technical replicates within each biological replicate. Statistical significance of the difference in gene expression between the 4.0 h treatment and 5.5 h treatment from 5.0 h to 8.0 h is shown on the top right of each plot. Statistical significance of the differences in gene expression at a specific time between the two conditions is shown on top of each point. (*$P < 0.05$; **$P < 0.01$; ***$P < 0.001$; ****$P < 0.0001$; repeated measures ANOVA followed by multiple comparisons. The subject factor, not the time factor, of the repeated measures ANOVA is reported here. Exact $P$ values provided in Table EV2.)

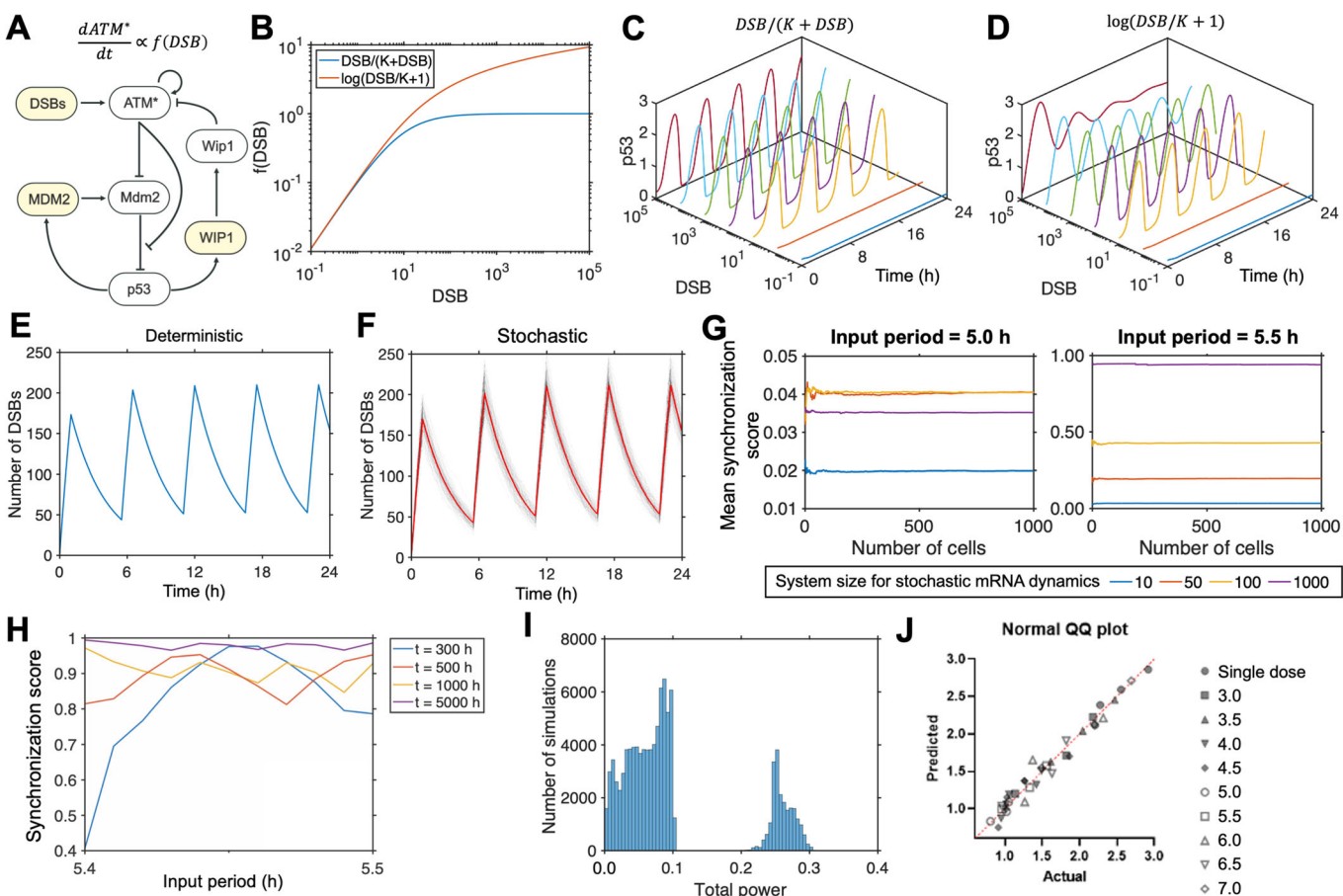

**Figure EV6.   Modifications to the mathematical model and statistical considerations.**

(A) Schematic of the p53 DSB response network where the rate of change in ATM* is directly proportional to a DSB input function. (B) Plots of the saturable and non-saturable DSB input functions for different values of DSBs. (C, D) Simulations of the p53 system with the saturable (C) and non-saturable (D) DSB input functions for different values of DSBs showing similar behavior at low damage but divergent behavior at high damage. (E) Deterministic DSB induction and repair with NCS-induced damage being applied every 5.5 h with a NCS-induced birth rate of 200 breaks/h, showing a characteristic sawtooth shape. (F) Stochastic DSB inductions and repair with same conditions as (E). Gray lines represent individual realizations of the stochastic process; the red line represents the mean across multiple runs. (G) Plot showing the average synchronization score as a function of the number of stochastic realizations for a non-entraining and an entraining input period under different levels of noise. Each plot shows convergence well before 1000 instances. (H) Synchronization score of the fully deterministic system for different simulation lengths showing fluctuations in shorter simulations due to the finite numerical nature of the DFT used to calculate the synchronization score. (I) Histogram of the total spectral power at steady state for systems with parameter sets randomly sampled in a twofold range around the basal values and simulated with 0 to 1000 DSBs, in intervals of 20, as the input. (J) QQ plot of the Shapiro–Wilk normality test for data in Figs. 1 and 2.

