## [Peer Review File · Molecular Systems Biology]

Pulsed stimuli enable p53 phase resetting to synchronize single cells and modulate cell-fate

Harish Venkatachalapathy, Samuel Dallon, Zhilin Yang, Samira Azarin, Casim Sarkar, and Eric Batchelor

Corresponding author(s): Eric Batchelor (ebatchel@umn.edu) , Casim Sarkar (csarkar@umn.edu)

Review Timeline:

Submission Date:	23rd Oct 23
Editorial Decision:	26th Oct 23
Appeal Received:	4th Mar 24
Editorial Decision:	3rd Apr 24
Revision Received:	22nd Nov 24
Editorial Decision:	18th Dec 24
Revision Received:	3rd Feb 25
Accepted:	10th Feb 25

Editors: Maria Polychronidou and Poonam Bheda

Transaction Report:

26th Oct 2023

Manuscript Number: MSB-2023-12077

Dear Eric,

Thank you for submitting your manuscript "Pulsed stimuli entrain p53 to synchronize single cells and modulate cell-fate determination" to Molecular Systems Biology.

We have now considered your manuscript and I regret to inform you that we have decided to not send it out for peer review.

In this study, you examine whether periodic DNA double-strand break (DSB) induction can increase cell-to-cell synchrony in the p53 response by entrainment. Using a mathematical model you predict entrainment conditions and then experimentally observe entrainment over a wider range of DSB frequencies than those predicted by the model. We appreciate the finding that this increased entrainment range can be explained by a p53 DSB response model exhibiting less robust oscillations. We also acknowledge that you report that p53 entrainment can affect target gene expression dynamics, depending on the stability of the target mRNA. While we appreciate that the presented findings may be relevant for guiding follow up investigations, we feel that as it stands the potential applications for achieving desired cell fate outcomes and the relevance in the context of tumors remains to be further explored. Overall, we are not convinced that the study provides the kind of decisive advance and demonstrated relevance for cancer therapy development that would be required for publication in Molecular Systems Biology.

That said, your work is a good candidate for Life Science Alliance (<http://www.life-science-alliance.org/>) our broad scope Open Access journal published in partnership between EMBO Press, Rockefeller University Press, and Cold Spring Harbor Laboratory Press. The editors of Life Science Alliance would be pleased to send your manuscript for peer review. Please use the following link to transfer your manuscript (no reformatting is required): Link Not Available.

Eric Sawey, executive editor of Life Science Alliance (e.sawey@life-science-alliance.org), would be happy to answer any questions.

I apologize for not bringing better news regarding the publication of your study in Molecular Systems Biology and I hope that you will view the possibility of a transfer to Life Science Alliance favorably.

Kind regards,

Maria

Maria Polychronidou, PhD
Senior Editor
Molecular Systems Biology

** As a service to authors, EMBO Press offers the possibility to directly transfer declined manuscripts to another EMBO Press title or to the open access journal Life Science Alliance launched in partnership between EMBO Press, Rockefeller University Press and Cold Spring Harbor Laboratory Press. The full manuscript and if applicable, reviewers' reports, are automatically sent to the receiving journal to allow for fast handling and a prompt decision on your manuscript. For more details of this service, and to transfer your manuscript please click on Link Not Available. **

UNIVERSITY OF MINNESOTA

Twin Cities Campus

**Department of Integrative Biology
and Physiology**

Medical School

*Eric Batchelor, Ph.D.
CCRB 3-136
2231 6th St SE
Minneapolis, MN 55455**Office: 612-301-3104
Fax: 612-301-1543**Email: ebatchel@umn.edu*

4 March 2024

Dear Dr. Polychronidou,

We would like to submit our enclosed manuscript, “**Pulsed stimuli entrain p53 to synchronize single cells and modulate cell-fate determination**,” for publication in *Molecular Systems Biology*. The manuscript describes our recent work detailing how the temporal dynamics of p53 expression can be entrained to reduce heterogeneity in cell signaling states across individual cells, impacting downstream stress responses.

Heterogeneity in the responses of individual cells to a stress agent is a broadly important biological phenomenon. In some contexts, such as the sporulation decision in bacteria, it provides a mechanism to enhance a favorable survival outcome for the bacterial population as a whole. However, in other biological contexts, including clinical responses to cancer chemotherapeutics, heterogenous responses in the form of fractional cell killing are a major hurdle for therapy. A key regulator of cell stress responses in human cells is the transcription factor p53. Work from our lab and others have shown that variability in p53 expression dynamics are important factors in determining diverse cellular outcomes to DNA damaging stresses in single cells (Paek et al *Cell* 2016; Reyes et al *Mol Cell* 2018; Hanson and Batchelor *Mol Syst Biol* 2022). In this study, we sought to take a complementary approach – can variability in p53 dynamics be reduced in a controllable manner? And, consequently, does reduction in p53 expression variability reduce heterogeneity in the activation of downstream pathways related to cell fate control?

We determined p53 expression dynamics can be entrained to an external and controlled cue, similar to how oscillatory circadian dynamics are entrainable to the light-dark cycle. Using a differential equation-based modeling approach, we first predicted that the p53 system is theoretically entrainable, and we identified the putative range of entrainment stimulus frequencies for p53 dynamics. Using time-lapse microscopy, we experimentally validated our prediction, and surprisingly determined that the range of entrainment frequencies for the system is even greater than predicted by our model. Refinement of our theoretical analysis through parameter sensitivity analysis identified a general principle of biological entrainment: greater entrainment is possible in less robust biological oscillators. Thus, while p53 dynamics are inherently variable, the heterogeneity enhances its capacity for entrainment. We further showed experimentally that entrainment of p53 propagates downstream, leading to a reduction in the variability of target gene expression and the ability to qualitatively alter target expression profiles as a function of target mRNA decay rates relative to the p53 entrainment frequency. Furthermore, we directly measured an impact on the fidelity of cell cycle arrest as a function of the entrainment frequency.

Our study will be of interest to a wide range of researchers in the fields of signal transduction, cancer biology, and dynamical systems. Our work provides novel insight into the functioning of a key stress response regulator for human health, the transcription factor p53. Additionally, this study identifies a general principle for other oscillatory systems, not only those that have been shown to be

entrainable, including circadian oscillators and the NF- κ B system, but potentially other biological oscillators such as p53 that have not previously been identified as entrainable.

Thank you very much for considering this manuscript.

Sincerely,

Eric Batchelor, PhD

Assistant Professor, Dept of Integrative Biology and Physiology

University of Minnesota

ebatchel@umn.edu

3rd Apr 2024

Manuscript Number: MSB-2023-12077R-Q

Title: Pulsed stimuli entrain p53 to synchronize single cells and modulate cell-fate determination

Dear Eric,

Thank you again for submitting your work to Molecular Systems Biology. We have now heard back from the three reviewers who agreed to evaluate your study. As you will see below, the reviewers acknowledge that the study seems potentially interesting. However, they point out that as it stands the study seems somewhat preliminary. They mention that additional analyses are required to better support the main conclusions and enhance the impact of the study. We would invite you to address the issues raised in a major revision.

Without repeating all the comments listed below, some of the more fundamental issues are the following.

- The reported p53 entrainment needs to be better supported.
- The effect of p53 entrainment on cell fates needs to be further explored. This is mentioned by both reviewers #1 and #3 and echoes our editorial concerns before sending the study out for review.
- The findings related to cell cycle entry modulation (e.g. Figure 4D) need to be better explained and supported.
- Some level of mechanistic insight linking the observed experimental results and the model predictions would enhance the impact of the study.

All issues raised by the reviewers would need to be satisfactorily addressed. As you may already know, our editorial policy allows in principle a single round of major revision. It is therefore essential to provide responses to the reviewers' comments that are as complete as possible. If you have any questions or if you would like to discuss your revision plan with me, please feel free to get in touch.

On a more editorial level, we would ask you to address the following points:

- We would recommend providing the code in a format that does not require MATLAB.
- Please provide a .doc version of the manuscript text (including legends for main Figures and EV Figures) and individual production quality figure files for the main Figures and EV Figures (one file per figure).
- Please include 5 keywords.
- We have replaced Supplementary Information by the Expanded View (EV format). In this case (unless the number of EV figures becomes > 6 during revision), all additional figures can be provided as EV Figures. Please provide one file per EV Figure. Their legends should be included in the manuscript text. For detailed instructions regarding expanded view please refer to our Author Guidelines: .
- Table S1 should be provided as Table EV1. Please include the description of the table within the file itself.
- Please provide a "standfirst text" summarizing the study in one or two sentences (approximately 250 characters), three to four "bullet points" highlighting the main findings and a "synopsis image" (exactly 550px width and max 400px height, jpeg or png format) to highlight the paper on our homepage.
- All Materials and Methods need to be described in the main text. We would encourage you to use 'Structured Methods', our new Materials and Methods format. According to this format, the Material and Methods section should include a Reagents and Tools Table (listing key reagents, experimental models, software and relevant equipment and including their sources and relevant identifiers) followed by a Methods and Protocols section in which we encourage the authors to describe their methods using a step-by-step protocol format with bullet points, to facilitate the adoption of the methodologies across labs. More information on how to adhere to this format as well as downloadable templates (.doc or .xls) for the Reagents and Tools Table can be found in our author guidelines: . An example of a Method paper with Structured Methods can be found here:
- Please include a Data availability section describing how the data and code have been made available. This section needs to be formatted according to the example below:
The datasets and computer code produced in this study are available in the following databases:
 - Chip-Seq data: Gene Expression Omnibus GSE46748 (<https://www.ncbi.nlm.nih.gov/geo/query/acc.cgi?acc=GSE46748>)
 - Modeling computer scripts: GitHub (<https://github.com/SysBioChalmers/GECKO/releases/tag/v1.0>)
 - [data type]: [full name of the resource] [accession number/identifier] ([doi or URL or identifiers.org/DATABASE:ACCESSION])

- For data quantification: please specify the name of the statistical test used to generate error bars and P values, the number (n) of independent experiments (specify technical or biological replicates) underlying each data point and the test used to calculate p-values in each figure legend. The figure legends should contain a basic description of n, P and the test applied. Graphs must include a description of the bars and the error bars (s.d., s.e.m.).
 - Please include a "Disclosure & Competing Interests Statement" in the main text.
 - The References should be formatted according to the Molecular Systems Biology reference style (i.e., ordered alphabetically and listing the first 10 authors followed by et al).
 - When you resubmit your manuscript, please download our CHECKLIST (<https://bit.ly/EMBOPressAuthorChecklist>) and include the completed form in your submission.
- *Please note* that the Author Checklist will be published alongside the paper as part of the transparent process (<https://www.embopress.org/page/journal/17444292/authorguide#transparentprocess>).

If you feel you can satisfactorily deal with these points and those listed by the referees, you may wish to submit a revised version of your manuscript. Please attach a covering letter giving details of the way in which you have handled each of the points raised by the referees. A revised manuscript will be once again subject to review and you probably understand that we can give you no guarantee at this stage that the eventual outcome will be favorable.

Kind regards,

Maria

Maria Polychronidou, PhD
Senior Editor
Molecular Systems Biology

We realize that it is difficult to revise to a specific deadline. In the interest of protecting the conceptual advance provided by the work, we recommend a revision within 3 months (2nd Jul 2024). Please discuss the revision progress ahead of this time with the editor if you require more time to complete the revisions. Use the link below to submit your revision:

IMPORTANT: When you send your revision, we will require the following items:

1. the manuscript text in LaTeX, RTF or MS Word format
2. a letter with a detailed description of the changes made in response to the referees. Please specify clearly the exact places in the text (pages and paragraphs) where each change has been made in response to each specific comment given
3. three to four 'bullet points' highlighting the main findings of your study
4. a short 'blurb' text summarizing in two sentences the study (max. 250 characters)
5. a 'thumbnail image' (550px width and max 400px height, Illustrator, PowerPoint or jpeg format), which can be used as 'visual title' for the synopsis section of your paper.
6. Please include an author contributions statement after the Acknowledgements section (see <https://www.embopress.org/page/journal/17444292/authorguide>)
7. Please complete the CHECKLIST available at (<https://bit.ly/EMBOPressAuthorChecklist>). Please note that the Author Checklist will be published alongside the paper as part of the transparent process (<https://www.embopress.org/page/journal/17444292/authorguide#transparentprocess>).
8. When assembling figures, please refer to our figure preparation guideline in order to ensure proper formatting and readability in print as well as on screen:
<https://bit.ly/EMBOPressFigurePreparationGuideline>
See also figure legend guidelines: <https://www.embopress.org/page/journal/17444292/authorguide#figureformat>
9. Please note that corresponding authors are required to supply an ORCID ID for their name upon submission of a revised manuscript (EMBO Press signed a joint statement to encourage ORCID adoption). (<https://www.embopress.org/page/journal/17444292/authorguide#editorialprocess>)
Currently, our records indicate that the ORCID for your account is 0000-0003-3870-5615.

Link Not Available

*** PLEASE NOTE *** As part of the EMBO Press transparent editorial process initiative (see our Editorial at <https://dx.doi.org/10.1038/msb.2010.72>), Molecular Systems Biology publishes online a Review Process File with each accepted manuscripts. This file will be published in conjunction with your paper and will include the anonymous referee reports, your point-by-point response and all pertinent correspondence relating to the manuscript. If you do NOT want this File to be published, please inform the editorial office at msb@embo.org within 14 days upon receipt of the present letter.

Reviewer #1:

In this study, Venkatachalapathy et al. asked whether a cellular response without the need for coherency across cells can still be entrained. To this end, they investigated if p53 oscillations after DNA DSBs can be entrained by a repetitive treatment of NCS, an antibiotic drug that introduces DNA DSBs. Through experiments and mathematical modeling, they showed that a repetitive NCS treatment could synchronize p53 period in a cell population to be closer to the time interval of the NCS treatment, a phenomenon they regarded entrainment. Further, the bandwidth of entrainable period observed from experiments was wider compared to that that in simulations. Using computer simulations, they demonstrated that a less rigid oscillator could have a wider bandwidth of entrainable period. Finally, they showed that entrainment of p53 oscillator altered the expression of its downstream targets depending on their mRNA half-lives. This study provides experimental and modeling results to indicate an interesting possibility of p53 as a weak but versatile oscillator that can be better responsive and entrained by external, repetitive signals. Nevertheless, overall, there is a need for stronger mechanistic- and system-level evidence to support their major arguments for p53 entrainment. Please see the following for my major and minor concerns.

Major concerns:

1. Based on the reference the author cited, "entrainment is a phenomenon in which two oscillators interact with each other, typically through physical or chemical means, to synchronize their oscillations." To introduce entrainment of p53 oscillations, they applied two NCS treatments with desired time interval (i.e., period) and observed higher synchrony in p53 oscillations in a cell population. Despite the higher degree of synchrony they observed, a more thorough and complete characterization with a different experimental design is needed to claim p53 entrainment. In biology, especially in the field of chronobiology, the tests of entrainment usually involve 1. Introduction of a long duration of repetitive stimulus and 2. Examination of oscillation period after withdrawing the stimulus. When the oscillators are entrained by external periodic stimuli, their periods will remain the period of stimuli for a period of time before they gradually return back to their intrinsic, free-running periods. To strengthen their observations, I would consider to (A) compare NCS treatments of different repeat numbers (e.g., from 2 to 5 repeats) with a period of 5.5 h and quantify the degree of entrainment via the level of synchrony in p53 period for Figure. 1; (B) Use the ideal treatment scheme (the number of repeats, ideally > 2, with a high level of entrainment) identified from (A) but with different periods to examine the likelihood of entrainment by different periods of NCS treatment for Figure 2.
2. The quantification of entrainability is different across the study. In simulations, the entrainment score is to compare the p53 oscillation period with the input NCS treatment period. However, in Figure 2B, the entrainment metric is the ratio of median absolute deviations of the second and first pulse timing. This measurement is more about the synchrony of p53 across cells instead of between NCS treatment and p53 oscillation. The definition of entrainment is better unified across the study.
3. There is a lack of mechanistic link between their experimental results and the simulations of their mechanistic model. They claimed that a weaker oscillator (e.g., with higher extrinsic noise) appeared to have a shallower valley at the limit cycle. What is the major biochemical mechanism in the p53 oscillator that can make it to be a weaker oscillator? Can this model-inspired mechanism be experimentally tested to make the p53 oscillator to be more rigid and less entrainable?
4. In the example provided in Figure 3E, the first valley of the two curves aren't really different. Is there a mechanistic explanation for this?
5. In Figure 4D, the authors stated that a second NCS dose at 5.5 hour would lead to more cell cycle reentries before 8th hour using p21 dynamics as a marker. However, in the single cell traces of second NCS dose at 5.5 hour, among all the cells, there are only 3 cells exhibiting an earlier decrease in p21 level. How robust is this result? To make this conclusion stronger, there is a need for (A) reporting the number of cells analyzed; (B) multiple biological repeats; (C) statistical analyses. Ideally, there is a parallel, direct measurement of CDK2 activity (e.g., CDK2 activity reporter or IF).
6. To strengthen their argument that p53 entrainment can modulate cell fate, the authors should consider examining the differential cell fate decision with optimal entrainment scheme (e.g., more repeats of NCS) suggested in point 1.
7. As mentioned by the authors, the biological significance of the entrainable p53 oscillator remains unclear. Considering biological systems need to maintain existing functions and remain adaptive and versatile in response to perturbations, the authors may want to leverage their modeling approach to indicate and discuss the possible mechanisms and functions for the weakness/evolvability of the p53 system.

Minor concerns

1. Figure 2B, y axis should be $MAD(2nd\ peak\ timing)/MAD(1st\ peak\ timing)$.
2. Figure EV2, please add the time axis and specify the cell numbers at the x and y axis of every panels.
3. Figure 2B, please indicate the number of quantified cells in each replicate.
4. In page 11, Figure 4D and 4E in the text should be 3D and 3E, respectively.
5. In Figure 4E and 4G, the statistic applied here need to be reconsidered. Repeated ANOVA is used when the subjects are repeatedly measured at different conditions or time points, and is used to test if different conditions or time points have distinct mean value. As a result, it can only show if the RT-qPCR signals are different between different time points, but not if the RT-qPCR signals across time points are different between two treatments.

Reviewer #2:

In the manuscript from Venkatachalapathy and colleagues they find that p53 oscillations can be entrained to a wide range of stimulus periods, more than is expected based on pre-published math models. Altering the math models in a way that increases the entrainment range results in a weaker oscillator. They further show that different pulse frequencies of NCS treatment result in differences in gene expression patterns for genes with short half-lives. Overall this manuscript should be of great interest to researchers studying p53 dynamics and dynamical systems. There are some minor points to address before it is fit for publication but after addressing these points I believe this manuscript would be an excellent fit for Molecular Systems Biology.

1. The authors use two different metrics for entrainment based on whether they are analyzing simulations (Entrainment Score) or actual data ($MAD\ 2nd\ peak / MAD\ 1st\ peak$). These two metrics reflect different things, the entrainment score near one suggests the system has the same frequency as the input. While the $MAD\ 2nd / MAD\ 1st$ metric near one shows that the 2nd peak shows the same variation as the first peak. Is there a way to measure the frequency of the experimental data to show that it matches the input frequency? I think this is important for figure 2 especially as the central argument of the paper is that p53 oscillations can be entrained to different input frequencies of DNA DSBs.
2. Figure 1F plots the MAD of 2nd peak/ MAD of 1st peak. In figure 2B the y-axis and legend show this as $MAD\ 1st/ MAD\ 2nd$. I believe 1F is correct?
3. Figure 2/ Figure EV2. I personally find the heat maps more instructive than the plots in Figure 2A. This is just a suggestion but I think it would be better to use the heat maps in Figure EV2 for Figure 2, and sort the cells based on the timing of the 2nd peak of p53. In this way the reader can visualize the distribution of timing of the 2nd peak. Again just a suggestion, as it is certainly possible this will not look as clear as I think.
4. For Figure 2 and Figure EV2 - It would be nice to add an indicator (dashed vertical line or carrot) on the graph of exactly when the second NCS dose was added.
5. Figure EV2, the authors should label the timepoints on the x-axis.
6. Figure 3A. I think it is important to outline what is going on here either in the figure legend or in the results section of the manuscript so it is more clear to readers.
7. For the more entrainable model in Figure 3, I think the authors should show some of the data produced by the model. It is not clear from the manuscript if this new sloppy model recapitulates some of the observations shown earlier. Showing that the model is closer to the data shown in Figure 2 would be nice. It would also be good to show that the sloppy model maintains other key aspects of the p53 math model like the 5.5 hour period.
8. Figure 3E: What is meant by U? No definition of U or Potential is provided in the manuscript.
9. On page 11 the authors reference Figures 4D and 4E. I think they mean 3D and 3E.
10. The first time NCS is used the authors should spell out neocarzinostatin.

Reviewer #3:

Venkatachalapathy and colleagues study the entrainment of p53 oscillations in response to double-stranded DNA breaks (DSBs). They demonstrate with experiments on MCF7 cells that the second pulse of DSB induction evokes a narrower distribution of p53 oscillation timings, albeit with a smaller average amplitude.

The paper is written well, the presentation is clear with clean plots and solid modelling and statistical quantification. The supplementary code and data are well described, although, having no access to MATLAB, I could not run the code myself. However, there are several issues that I'd like the authors to discuss further.

1. My main critique pertains to the entire study based only on two pulses of DSB induction, which is then used to justify the entrainment of the p53 oscillatory network. Indeed, the distribution of the 2nd peak in the double dose experiment is narrower, but how long this effect lasts is unclear. Already the 2nd peak has a lower average amplitude. Perhaps the "entrainment" is lost in the long term?

The authors suggest in the discussion that the entrainment could be a way to synchronize cells to make them more responsive

to a secondary treatment. This is an attractive proposition, but based on demonstrated results, especially those in Figure 4 where the authors study the effect on downstream p53 targets, the phenotypic effect of the entrainment seems rather weak. Therefore, despite the thoroughness of the analysis, the impact of this study is unconvincing.

2. Even though the timing distribution of the second peak of p53-mVenus expression is narrower (Fig. 1 E & F), its average amplitude is lower (Fig. 1 D, Fig. 2). The amplitude decrease seems also evident at the single-cell level (Fig. 2). How would it look like long-term? Is the "entrainment" maintained long-term, since the amplitude seems to fade? Please quantify the amplitude of the 1st vs 2nd peak.

How are these experimental results reconciled with the model, which seems to predict sustained oscillations (Fig.3)?

3. Fig. 4D, right panel, shows only 3 traces with rapid p21 degradation. Does it mean that only these 3 cells re-enter the cell cycle? What about magenta traces that exhibit sharp decrease right after the green period of 5-8h? I find the explanation of this intriguing phenomenon in the text a little too scarce. The modulation of the cell cycle entry due to p53 dynamics would be worthwhile to pursue by using a CDK2 biosensor (Spencer et al. 2013), and would further strengthen the manuscript.

Also, it would be interesting to see how the fraction of cells with this rapid p21 decrease changed for other double dose regimes, both with a longer and shorter period between dosages.

Minor:

1. Page 11. The reference to Figure 4D and E should be referring to Fig. 3 instead.
2. I haven't found this explicitly explained in the Methods; how exactly was the double pulse of NCS delivered into wells?

Response to Reviewers' Comments

We thank the reviewers for their thorough and insightful reviews of our initial manuscript. We truly appreciate their time and effort in the review process. We have worked hard to address the concerns that they raised. By addressing their concerns, we feel that the manuscript has been greatly improved in both the impact and the clarity of our findings. Below, please find a point-by-point discussion of how we have addressed their concerns, and we hope that the reviewers and editors may find the manuscript suitable for publication in *Molecular Systems Biology*.

Sincerely,

Eric Batchelor

Reviewer #1:

Major concerns:

1. Based on the reference the author cited, "entrainment is a phenomenon in which two oscillators interact with each other, typically through physical or chemical means, to synchronize their oscillations." To introduce entrainment of p53 oscillations, they applied two NCS treatments with desired time interval (i.e., period) and observed higher synchrony in p53 oscillations in a cell population. Despite the higher degree of synchrony they observed, a more thorough and complete characterization with a different experimental design is needed to claim p53 entrainment. In biology, especially in the field of chronobiology, the tests of entrainment usually involve 1. Introduction of a long duration of repetitive stimulus and 2. Examination of oscillation period after withdrawing the stimulus. When the oscillators are entrained by external periodic stimuli, their periods will remain the period of stimuli for a period of time before they gradually return back to their intrinsic, free-running periods. To strengthen their observations, I would consider to (A) compare NCS treatments of different repeat numbers (e.g., from 2 to 5 repeats) with a period of 5.5 h and quantify the degree of entrainment via the level of synchrony in p53 period for Figure. 1; (B) Use the ideal treatment scheme (the number of repeats, ideally > 2, with a high level of entrainment) identified from (A) but with different periods to examine the likelihood of entrainment by different periods of NCS treatment for Figure 2.

The reviewer has raised many excellent points in this comment, which are echoed by concerns raised by Reviewer 3 regarding better characterization and testing of the nature of increased p53 synchronization when driven by a repeated external stimulus. We have performed several

experiments to better elucidate the phenomenon that we observed in our original manuscript. As suggested, we moved beyond 2 doses of NCS to also look at the p53 response to 3, 4, and 5 repeated doses of NCS with a 5.5-h period between doses. Furthermore, we quantified the synchrony in the p53 response upon cessation of these greater numbers of external stimulation. As shown in the new Figure EV2D, the synchronization in the p53 response was indeed present for the increased number of NCS doses. Furthermore, we looked at a series of 5 doses of NCS at the lower range of synchronization, a 4-h period, as indicated in the new Fig. EV2D. Again, increased synchrony in p53 pulses driven at the 4-h period was also observed for the course of 5 doses. These results indicate that the increased synchronization of p53 pulses could be maintained longer than the 2-dose duration that we originally observed.

As suggested by the reviewer, we also analyzed the effect of p53 synchronization upon removal of the external NCS stimulation. We first generated a prediction based on our computational model. As show in new Fig EV2A, our model predicted that the p53 dynamics would rapidly return to the natural p53 pulse frequency upon removal of the external driving stimulus. We quantified the interpeak timing between the two externally driven p53 pulses from our initial double dose experiments at a 5.5-h or 4-h period, and compared those values to the interpeak timing between the last externally driven pulse and a subsequent third pulse (new Fig. EV2B-C). As predicted by our model, the 4-h driven pulses rapidly returned to a longer 5.5-h period upon cessation of the NCS doses. This rapid loss of synchrony was also observed in the experiments in Fig. EV2D-E upon cessation of the external stimulation.

In light of these new computational and experimental results, we now in this revised manuscript no longer refer to the observed increase in synchronization as entrainment. In consultation with experts in the field of chronobiology, we now more accurately identify the phenomenon as “phase resetting.” We have made substantial and appropriate changes to the manuscript title, text, and figures to more accurately report our findings. We are truly grateful for the reviewers’ comments on this aspect of our original manuscript, as it has led us to more carefully analyze this system and more precisely identify and characterize the novel result.

2. The quantification of entrainability is different across the study. In simulations, the entrainment score is to compare the p53 oscillation period with the input NCS treatment period. However, in Figure 2B, the entrainment metric is the ratio of median absolute deviations of the second and first pulse timing. This measurement is more about the synchrony of p53 across cells instead of between NCS treatment and p53 oscillation. The definition of entrainment is better unified across the study.

Given our reassessment of the reported phenomenon as described in our response to Major Comment 1, throughout the revised manuscript we now report the synchrony of p53 pulse timing across cells and between individual pulses, rather than report entrainment of p53 to the external NCS doses.

The difference in criteria is due to the difference in time scales that can be evaluated computationally and experimentally. Computational simulations showed that it could take ~10 period times (~55 hours) for the system frequency to equal the input frequency (EV2A). Thus, to feasibly evaluate this phenomenon in a reasonable experimental time scale, we instead utilized the fact that strong phase resetting would be able to overcome noise within the system and result in increased synchrony. We now include a description of the motivation for these two synchronization criteria in the main text.

3. There is a lack of mechanistic link between their experimental results and the simulations of their mechanistic model. They claimed that a weaker oscillator (e.g., with higher extrinsic noise) appeared to have a shallower valley at the limit cycle. What is the major biochemical mechanism in the p53 oscillator that can make it to be a weaker oscillator? Can this model-inspired mechanism be experimentally tested to make the p53 oscillator to be more rigid and less entrainable?

Identifying specific biochemical parameters that give rise to complex dynamical behavior can be a rather challenging task, especially for nonlinear systems such as the p53 network. While attempting to address this concern, our analysis of the parameters in our computational model showed that sloppiness can arise from changing most of the parameters in the system, so it is likely not feasible to identify any given parameter that could generate the desired perturbation experimentally to alter the rigidity of the p53 oscillator. This difficulty is also compounded by the nature of the feedbacks in the network, as there are compensatory effects that can counteract perturbations to targeted parameters in our reduced model. Additionally, due to the interconnectedness of the p53 network, perturbation of many of the parameters can often have pleiotropic effects, including cell lethality.

To attempt to address this concern, we considered theoretically and computationally why a sloppier system would be evolutionarily favorable (which is connected to Comment 7 below). Typically, one would expect that a sloppier oscillator makes the transition to a sustained state easier in cases of a sudden change in stimulus (for example, additional DNA damage). This is indeed the case when we simulate a sloppier oscillator compared with a more rigid oscillator – the oscillatory transients decay significantly faster for a sloppy system when the system approaches a fixed-point stable steady state (new Fig. EV3B). We have included this new computational analysis in the revised manuscript, and we have also expanded our discussion of sloppy versus rigid oscillators in the main text.

4. In the example provided in Figure 3E, the first valley of the two curves aren't really different. Is there a mechanistic explanation for this?

It has been previously reported that positive feedback in specific phases of the oscillations are linked to increased robustness or decreased noise in those regions (Gerard et al. *FEBS J* 2012 doi: 10.1111/j.1742-4658.2012.08585.x). Here, both scenarios being evaluated involve strong self-activation of ATM, which acts primarily at the beginning of the oscillations during which there are relatively low levels of p53 and Mdm2. Since the positive feedback parameter A (self-activation constant of ATM) is high in both landscapes, this results in low noise in the region of the first valley (low p53, low Mdm2) in both cases. Examining the steady state traces of stochastic simulations below, we can see that the traces band together significantly near the low p53, low Mdm2 region (left, middle panels). In contrast, weakening this parameter results in increased noise (and thus a wider valley) at the low p53, low Mdm2 region (right panel).

5. In Figure 4D, the authors stated that a second NCS dose at 5.5 hour would lead to more cell cycle reentries before 8th hour using p21 dynamics as a marker. However, in the single cell traces of second NCS dose at 5.5 hour, among all the cells, there are only 3 cells exhibiting an earlier decrease in p21 level. How robust is this result? To make this conclusion stronger, there is a need for (A) reporting the number of cells analyzed; (B) multiple biological repeats; (C) statistical analyses. Ideally, there is a parallel, direct measurement of CDK2 activity (e.g., CDK2 activity reporter or IF).

To better characterize the loss of cell cycle arrest suggested by the rare cell events reported in the original manuscript, we clarified our initial experimental conditions as well as performed additional experiments. We now explicitly show all traces from the single dose and double dose conditions (Fig. EV4A-B), as well as explicitly reported the number of cells ($n = 121, 135, 148$ cells; Fig. 4A caption) from the experiments performed in biological triplicate.

To further validate the result of rare cell escape from cell cycle arrest, we extended the duration of synchronization to three pulses of NCS in both the 4-h and 5.5-h repeated dosing conditions

and quantified the number of escaper cells (new Fig. 4E-F). Again, the experiments were performed in biological triplicates, for which we analyzed a total of $n = 2162$ responding cells. These conditions resulted in a much larger number of escaper cells observed (new Fig. 4F). Statistical analysis by t-test showed a statistically significant 4.5-fold increase in the percentage of escaper cells between the 5.5-h period triple NCS treatment compared to the 4.0-h period triple NCS treatment (Fig. 4G).

Finally, as suggested by both Reviewers 1 and 3, we corroborated these results from the p21 live-cell reporter by using a previously published live-cell reporter of CDK2 activity (new Fig. EV4C-D).

6. To strengthen their argument that p53 entrainment can modulate cell fate, the authors should consider examining the differential cell fate decision with optimal entrainment scheme (e.g., more repeats of NCS) suggested in point 1.

Please see the response to the above comment, in which we have described our new experimental results in which we have also examined the fidelity of cell cycle arrest following three doses of NCS in addition to our original results with a double dose of NCS.

7. As mentioned by the authors, the biological significance of the entrainable p53 oscillator remains unclear. Considering biological systems need to maintain existing functions and remain adaptive and versatile in response to perturbations, the authors may want to leverage their modeling approach to indicate and discuss the possible mechanisms and functions for the weakness/evolvability of the p53 system.

While the entrainability of the p53 oscillator is no longer being considered in the revised manuscript, as discussed in response to Comment 3 above we can consider potential benefits of a sloppy versus rigid operator that are germane to this original Comment. Through analysis of our computational model, we find that a sloppier p53 oscillator can more quickly eliminate transients upon cessation of an external stimulus, returning to a steady state level. This could provide an evolutionary benefit for exiting from repair states, halting cell cycle arrest, or preventing apoptosis when a p53-stimulating damage is removed. The new computational results and a discussion of this point are now included in the revised manuscript.

Minor concerns

1. Figure 2B, y axis should be $MAD(2nd\ peak\ timing)/MAD(1st\ peak\ timing)$.

We have made the correction in the y-axis label.

2. Figure EV2, please add the time axis and specify the cell numbers at the x and y axis of every panels.

We have made the suggested changes.

3. Figure 2B, please indicate the number of quantified cells in each replicate.

We now indicate in the figure caption that each biological replicate contains quantification from a minimum of 45 individual cells, and all experiments were performed in at least biological triplicate.

4. In page 11, Figure 4D and 4E in the text should be 3D and 3E, respectively.

We apologize for the oversight, and we have made sure that figure references are correct in the current text.

5. In Figure 4E and 4G, the statistic applied here need to be reconsidered. Repeated ANOVA is used when the subjects are repeatedly measured at different conditions or time points, and is used to test if different conditions or time points have distinct mean value. As a result, it can only show if the RT-qPCR signals are different between different time points, but not if the RT-qPCR signals across time points are different between two treatments.

We apologize for the lack of clarity in reporting this specific result. We performed a repeated measures two-way ANOVA here which reports the statistical significance of two factors – one was the effect of time (as the reviewer mentions), the other was the effect of the treatment. The statistical significance of the treatment was reported in this manuscript. We have now clarified this in the manuscript.

Reviewer #2:

Overall this manuscript should be of great interest to researchers studying p53 dynamics and dynamical systems. There are some minor points to address before it is fit for publication but after addressing these points I believe this manuscript would be an excellent fit for Molecular Systems Biology.

We thank the reviewer for their interest in our findings and their support for publication.

1. The authors use two different metrics for entrainment based on whether they are analyzing simulations (Entrainment Score) or actual data (MAD 2nd peak / MAD 1st peak). These two metrics reflect different things, the entrainment score near one suggests the system has the same frequency as the input. While the MAD 2nd / MAD 1st metric near one shows that the 2nd peak shows the same variation as the first peak. Is there a way to measure the frequency of the experimental data to show that it matches the input frequency? I think this is important for figure 2 especially as the central argument of the paper is that p53 oscillations can be entrained to different input frequencies of DNA DSBs.

Reviewer 1 expressed a similar concern regarding the different metrics used for quantifying entrainability in the original manuscript (Major Comment 2 above). Our new analysis has indicated that the p53 system exhibits the phenomenon of phase resetting, rather than entrainment. Therefore, we have now made consistent use of a metric comparing the synchronization of pulses between cells, rather than entrainment to the external NCS stimulus.

2. Figure 1F plots the MAD of 2nd peak/ MAD of 1st peak. In figure 2B the y-axis and legend show this as MAD 1st/ MAD 2nd . I believe 1F is correct?

We again apologize for the oversight, which was also pointed out by Reviewer 1. We have made the correction in the Fig 2B.

3. Figure 2/ Figure EV2. I personally find the heat maps more instructive than the plots in Figure 2A. This is just a suggestion but I think it would be better to use the heat maps in Figure EV2 for Figure 2, and sort the cells based on the timing of the 2nd peak of p53. In this way the reader can visualize the distribution of timing of the 2nd peak. Again just a suggestion, as it is certainly possible this will not look as clear as I think.

Thanks for the suggestion. We agree that heat maps can be very informative and improve the clarity of single cell dynamic expression data. We now include the individual cell traces as well as heat maps for the experimental results.

4. For Figure 2 and Figure EV2 - It would be nice to add an indicator (dashed vertical line or carrot) on the graph of exactly when the second NCS dose was added.

Thanks also for this suggestion to improve the clarity of the data presentation. We have added lightning bolts to indicated when the external NCS treatments are applied.

5. Figure EV2, the authors should label the timepoints on the x-axis.

We have added the time points, as suggested.

6. Figure 3A. I think it is important to outline what is going on here either in the figure legend or in the results section of the manuscript so it is more clear to readers.

As suggested, we added more details in the Results section to better guide readers through the computational analysis outlined in this Figure.

7. For the more entrainable model in Figure 3, I think the authors should show some of the data produced by the model. It is not clear from the manuscript if this new sloppy model recapitulates some of the observations shown earlier. Showing that the model is closer to the data shown in Figure 2 would be nice. It would also be good to show that the sloppy model maintains other key aspects of the p53 math model like the 5.5 hour period.

We thank the reviewer for this suggestion. We have now included in the new Fig. EV3B modeling results showing p53 dynamics over a range of three systems, from more rigid to “sloppier” oscillators. From the model, we observe that for the different considered conditions, characteristics including the p53 pulse period are comparable. However, there is a striking difference in how quickly the p53 levels stop oscillating and return to a fixed steady state level upon cessation of the repeated external stimulus. We found that with a sloppier oscillator, there is a more rapid loss of the transient behavior, consistent with what we have observed experimentally and indicative of a phase resetting system rather than an entrained system. In addition to the modeling results in the new Figure, we have included a discussion of these results within the Main Text.

8. Figure 3E: What is meant by U? No definition of U or Potential is provided in the manuscript.

We apologize for this oversight. For clarity, we have now included a brief definition of what we mean by the potential U in addition to the cited references.

9. On page 11 the authors reference Figures 4D and 4E. I think they mean 3D and 3E.

We have checked our figure references for accuracy in the revised manuscript.

10. The first time NCS is used the authors should spell out neocarzinostatin.

We have made this correction.

Reviewer #3:

1. My main critique pertains to the entire study based only on two pulses of DSB induction, which is then used to justify the entrainment of the p53 oscillatory network. Indeed, the distribution of the 2nd peak in the double dose experiment is narrower, but how long this effect lasts is unclear. Already the 2nd peak has a lower average amplitude. Perhaps the "entrainment" is lost in the long term?

The authors suggest in the discussion that the entrainment could be a way to synchronize cells to make them more responsive to a secondary treatment. This is an attractive proposition, but based on demonstrated results, especially those in Figure 4 where the authors study the effect on downstream p53 targets, the phenotypic effect of the entrainment seems rather weak. Therefore, despite the thoroughness of the analysis, the impact of this study is unconvincing.

Similar concerns were expressed by Reviewer 1. As described in our response to Reviewer 1, Major Comment 1 above, we have performed extensive additional experiments and analysis to

better characterize the p53 system in response to additional doses of external stimulus beyond 2 doses, as well as analyzing the duration of the synchronizing effects upon cessation of external stimuli. We thank the Reviewers for their suggestions, which have been integral in helping clarify a major finding presented in our study.

2. Even though the timing distribution of the second peak of p53-mVenus expression is narrower (Fig.1 E & F), its average amplitude is lower (Fig. 1 D, Fig. 2). The amplitude decrease seems also evident at the single-cell level (Fig. 2). How would it look like long-term? Is the "entrainment" maintained long-term, since the amplitude seems to fade? Please quantify the amplitude of the 1st vs 2nd peak.

How are these experimental results reconciled with the model, which seems to predict sustained oscillations (Fig.3)?

We thank the Reviewer for their careful reading of our original manuscript. Experimentally, we indeed observed a reduction in amplitude, which is predicted by our model as well. The reduction is due to the upregulation of Mdm2 from its steady state levels upon DNA damage induction. However, with noisiness present in the real, experimental system, not all cells have this reduction in second pulse amplitude. Instead, there is a negative correlation between the amplitude of these two peaks (For the entrained region from 4.0 to 5.5 h, Spearman correlation = -0.4688, p-value < 0.0001). We now describe this observation and cite the appropriate reference in the revised manuscript.

3. Fig. 4D, right panel, shows only 3 traces with rapid p21 degradation. Does it mean that only these 3 cells re-renter the cell cycle? What about magenta traces that exhibit sharp decrease right after the green period of 5-8h? I find the explanation of this intriguing phenomenon in the text a little too scarce. The modulation of the cell cycle entry due to p53 dynamics would be worthwhile to pursue by using a CDK2 biosensor (Spencer et al. 2013), and would further strengthen the manuscript.

Also, it would be interesting to see how the fraction of cells with this rapid p21 decrease changed for other double dose regimes, both with a longer and shorter period between dosages.

Reviewer 1 raised many of the same issues and had many similar suggestions for improving our analysis of cell cycle arrest fidelity upon synchronization. As described in our response to Reviewer 1, Comment 5, we performed several additional experiments to elaborate on our initial observations, including: extending our analysis to 3, 4, and 5 repeated doses of NCS;

comparing the additional repeated doses under conditions of a 4-h period versus a 5.5-h period; and corroborating our findings from the p21 reporter with the suggested live-cell reporter of CDK2 activity. We have included these results in new subfigures Fig. 4E-G as well as a new Expanded View Figure EV4A-D.

Minor:

1. Page 11. The reference to Figure 4D and E should be referring to Fig. 3 instead.

We apologize for this oversight and we have checked all figure references in the revised manuscript.

2. I haven't found this explicitly explained in the Methods; how exactly was the double pulse of NCS delivered into wells?

It was delivered by manual addition to the wells. We now state this in the Methods section.

18th Dec 2024

Manuscript Number: MSB-2023-12077RR

Title: Pulsed stimuli enable p53 phase resetting to synchronize single cells and modulate cell-fate

Dear Dr Batchelor,

Thank you for the submission of your revised manuscript to Molecular Systems Biology. I am pleased to inform you that we will be able to accept your manuscript pending the following final amendments and appropriate response to reviewers:

- 1) Please check the "Author Checklist" carefully and complete all relevant questions. Currently information in the section on "Laboratory Protocol" is missing.
- 2) Author contributions: Please remove it from the manuscript and specify author contributions in our submission system. CRediT has replaced the traditional author contributions section because it offers a systematic machine-readable author contributions format that allows for more effective research assessment. You are encouraged to use the free text boxes beneath each contributing author's name to add specific details on the author's contribution. More information is available in our guide to authors:
<https://www.embopress.org/page/journal/17574684/authorguide#authorshipguidelines>
- 3) Our journal encourages inclusion of *data citations in the reference list* to directly cite datasets that were re-used and obtained from public databases. Data citations in the article text are distinct from normal bibliographical citations and should directly link to the database records from which the data can be accessed. In the main text, data citations are formatted as follows: "Data ref: Smith et al, 2001" or "Data ref: NCBI Sequence Read Archive PRJNA342805, 2017". In the Reference list, data citations must be labeled with "[DATASET]". A data reference must provide the database name, accession number/identifiers and a resolvable link to the landing page from which the data can be accessed at the end of the reference. Further instructions are available at .
- 4) In the Methods, please take care of the following:
 - The Materials and Methods section should be renamed to "Methods".
 - Cell lines: Please include an identifier or catalog number for HEK-293T in the Reagents and Tools Table.
 - Please also be sure to include a sentence in the Methods as to whether or not the cell lines were recently authenticated.
 - Please ensure that a statement on whether or not blinding was done is included in the Methods even if no blinding was done. Please also be sure to update the Author Checklist with this information and where it can be found in the manuscript.
- 5) For the figures and figure legends, please take care of the following:
 - Please make sure to update the callouts of all figures in the main manuscript text. Currently there is a callout for a Table 1 but we could only find a Table EV1; the callout for Figure EV5 seems to be missing.
 - Please note that the individual figure legends for figure EV 5a-d is not provided in the manuscript. This needs to be rectified.
 - Please note that the figure title for figures EV 1-6 is not provided in the manuscript. This needs to be rectified.
 - Please define the annotated p values */**** as well as provide the exact p-values for the same in the legend of figure 1f; 4c; as appropriate.
 - Please note that the exact p values are not provided in the legends of figures 1e; 2b; 4g; 5b, d; EV 2b; EV 4d; EV 5d.
 - Please indicate the statistical test used for data analysis in the legends of figures EV 2b-c; EV 5d.
 - Please note that in figures 5b, d; EV 5d; there is a mismatch between the annotated p values in the figure legend and the annotated p values in the figure file that should be corrected.
 - Please note that the box plots need to be defined in terms of minima, maxima, centre, bounds of box and whiskers, and percentile in the legend of figure 4c.
 - Please note that information related to n is missing in the legends of figures 4c; EV 2c.
 - Although 'n' is provided, please describe the nature of entity for 'n' in the legend of figure EV 1c.
- 6) Funding: Please note that funding information given in our submission system comments box includes University of Minnesota, which should be included in the Funders list in the system.
- 7) Synopsis: Please check your synopsis text and image before submission with your revised manuscript. Please be aware that in the proof stage minor corrections only are allowed (e.g., typos).
- 8) Source Data: Please ensure that a completed Source Data checklist is uploaded as a Related Manuscript File (this file will be sent to you by my colleague Hannah Sonntag). Source Data should be organized as a single source data file (zipped) per figure for main figures (all EV and/or Appendix figure Source Data can be included in a single folder), with the panels clearly visible in the folder structure instead of a single excel file for all Source Data. e.g. all the Source data files for figure 1 need to be saved in a single folder and this needs to be zipped and then uploaded as "SD figure 1.zip" file.
- 9) As part of the EMBO Publications transparent editorial process initiative (see our policy here: https://www.embopress.org/transparent-process#Review_Process), Molecular Systems Biology will publish online a Peer Review File (PRF) to accompany accepted manuscripts. This file will be published in conjunction with your paper and will include the anonymous referee reports, your point-by-point response and all pertinent correspondence relating to the manuscript. Let us know whether you agree with the publication of the PRF and as here, if you want to remove or not any figures from it prior to publication. Please note that the Authors checklist will be published at the end of the PRF.
- 10) Please provide a point-by-point letter INCLUDING my comments as well as the reviewer's reports and your detailed

responses (as Word file).

I look forward to reading a new revised version of your manuscript as soon as possible.

Yours sincerely,

Poonam Bheda, PhD
Scientific Editor
Molecular Systems Biology

Reviewer #1:

In this revised manuscript, the authors have addressed my primary concerns and major points. In particular, they have revised the description on synchronized p53 responses to "phase resetting" rather than "entrainment of p53 oscillations", with a more precise definition. I also appreciate the new experimental results where they tested the synchrony in the p53 response with up to 5 repeated doses of NCS. The inclusion of additional traces for p21 dynamics and CDK2 activity measurements further strengthens their argument that cell fate could be modulated by p53 phase resetting. I have a few minor questions and suggestions regarding this revised work:

1. The authors showed that pushing parameter A (self-activation of ATM) closer its right limit ($10^{0.22}$ fold change) of oscillatory regime led to a sloppier landscape (Fig. 3E). Beyond parameter A, have the authors systematically perturbed each parameter toward its right and left limits to examine the resulting landscapes? Does the statement about the sloppier landscape still hold under these conditions?
2. Could the authors clarify the value of parameter A in Fig. EV3B? Is it an exact value or a fold-change relative to the original value (30.5, as indicated in Table EV1)? Could the authors explain the rationale for selecting A to be 1.2~1.4? Are those values within the oscillatory regime?
3. The authors hypothesize that changes in p53 pulsing frequency, as a result of phase resetting, influence the downstream response (Figure 4). Could the authors elaborate and discuss how they deconvolute the specific effects of pulsing frequency from other dynamic features, such as amplitude, total signal (area under the curve), or additional temporal characteristics, on downstream target regulation?
4. Could the authors clarify how escaper cells are defined in Figure 4? Is the classification solely based on molecular characteristics (e.g., p21 level and/or Cdk2 activity) or does it also consider cellular phenotypes, such as cell division events?
5. To further enhance the significance of this study, we suggest the authors to expand the discussion on how different p53 phase resetting frequencies could generally impact the expression of its target genes with different mRNA half-lives, thereafter cellular phenotypes. For instance, as the authors have mentioned, genes with unstable mRNA may require higher p53 frequency to activate. In addition, to activate a gene with weaker p53-binding promoter may require lower frequency of p53, as p53 needs to maintain in a higher level for a longer time period to initiate the transcription. Furthermore, genes with longer mRNA half-life may behave closer to an integrator. Lower p53 frequencies could lead to a higher rate of accumulation of these genes.

Reviewer #2:

The authors have addressed all of my concerns and I feel the manuscript should be accepted for publication.

Reviewer #3:

The authors have addressed all of my concerns. The additional experiments strengthened the manuscript and the conclusions.

Dear Dr. Bheda,

We thank you and the reviewers for your reviews of our revised manuscript. We truly appreciate your time and effort in the review process. We have addressed the additional editorial concerns and the additional concerns of Reviewer #1 that were raised. We feel that the manuscript has been improved in clarity and broader impact. Below, please find a point-by-point discussion detailing how we have addressed the remaining concerns, and we hope that the reviewers and editors may find the manuscript suitable for publication in *Molecular Systems Biology*.

Sincerely,

Eric Batchelor

Editor's comments:

1) Please check the "Author Checklist" carefully and complete all relevant questions. Currently information in the section on "Laboratory Protocol" is missing.

We apologize for the oversight and we have corrected this error in the resubmitted Author Checklist.

2) Author contributions: Please remove it from the manuscript and specify author contributions in our submission system.

We have removed the section from the manuscript.

*3) Our journal encourages inclusion of *data citations in the reference list* to directly cite datasets that were re-used and obtained from public databases.*

No re-used, publicly available data has been used.

4) In the Methods, please take care of the following:

- The Materials and Methods section should be renamed to "Methods".*
- Cell lines: Please include an identifier or catalog number for HEK-293T in the Reagents and Tools Table.*
- Please also be sure to include a sentence in the Methods as to whether or not the cell lines were recently authenticated.*
- Please ensure that a statement on whether or not blinding was done is included in the Methods even if no blinding was done. Please also be sure to update the Author Checklist with this information and where it can be found in the manuscript.*

We have made all of the requested changes to the Methods section.

5) For the figures and figure legends, please take care of the following:

a) Please make sure to update the callouts of all figures in the main manuscript text. Currently there is a callout for a Table 1 but we could only find a Table EV1; the callout for Figure EV5 seems to be missing.

We have updated all callouts in the text.

b) Please note that the individual figure legends for figure EV 5a-d is not provided in the manuscript. This needs to be rectified.

The individual legends for EV5A-D have been included in the legend.

c) Please note that the figure title for figures EV 1-6 is not provided in the manuscript. This needs to be rectified.

We have added the EV legends to the manuscript main text.

*d) Please define the annotated p values */**** as well as provide the exact p-values for the same in the legend of figure 1f; 4c; as appropriate.*

and

e) Please note that the exact p values are not provided in the legends of figures 1e; 2b; 4g; 5b, d; EV 2b; EV 4d; EV 5d.

We have defined all annotated p-values in the appropriate legends. We also now include a new Table EV2 that lists the exact p-values for all reported statistical tests throughout the manuscript.

f) Please indicate the statistical test used for data analysis in the legends of figures EV 2b-c; EV 5d.

The statistical tests used have now been included in the figure legends.

g) Please note that in figures 5b, d; EV 5d; there is a mismatch between the annotated p values in the figure legend and the annotated p values in the figure file that should be corrected.

We apologize for the confusion, and we have ensured that the p-value symbols now match between the figure and the legend.

h) Please note that the box plots need to be defined in terms of minima, maxima, centre, bounds of box and whiskers, and percentile in the legend of figure 4c.

The whisker plots have been defined in the figure legend.

i) Please note that information related to n is missing in the legends of figures 4c; EV 2c.

The values for n have been included in the figure legends.

j) Although 'n' is provided, please describe the nature of entity for 'n' in the legend of figure EV 1c.

n has been defined in the figure legend as the number of stochastic simulations performed.

6) Funding: Please note that funding information given in our submission system comments box includes University of Minnesota, which should be included in the Funders list in the system.

We will include the University of Minnesota in the funders information in the system for our submission.

7) Synopsis: Please check your synopsis text and image before submission with your revised manuscript. Please be aware that in the proof stage minor corrections only are allowed (e.g., typos).

We have checked our submission.

8) Source Data: Please ensure that a completed Source Data checklist is uploaded as a Related Manuscript File (this file will be sent to you by my colleague Hannah Sonntag). Source Data should be organized as a single source data file (zipped) per figure for main figures (all EV and/or Appendix figure Source Data can be included in a single folder), with the panels clearly visible in the folder structure instead of a single excel file for all Source Data. e.g. all the Source data files for figure 1 need to be saved in a single folder and this needs to be zipped and then uploaded as "SD figure 1.zip" file.

Hannah Sonntag has not reached out to us yet, but we will be happy to provide the files upon her request.

9) As part of the EMBO Publications transparent editorial process initiative (see our policy here: https://www.embopress.org/transparent-process#Review_Process), Molecular Systems Biology will publish online a Peer Review File (PRF) to accompany accepted manuscripts. This file will be published in conjunction with your paper and will include the anonymous referee reports, your point-by-point response and all pertinent correspondence relating to the manuscript. Let us know whether you agree with the publication of the PRF and as here, if you want to remove or

not any figures from it prior to publication. Please note that the Authors checklist will be published at the end of the PRF.

We agree to the PRF being published.

Review 1 comments:

1. The authors showed that pushing parameter A (self-activation of ATM) closer its right limit ($10^{0.22}$ fold change) of oscillatory regime led to a sloppier landscape (Fig. 3E). Beyond parameter A, have the authors systematically perturbed each parameter toward its right and left limits to examine the resulting landscapes? Does the statement about the sloppier landscape still hold under these conditions?

While we did not perform an exhaustive perturbation of each parameter towards the left and right limits to examine all possible landscapes, we did find that the result remains qualitatively true for another important parameter – the Mdm2 degradation rate. Increasing this rate ($10^{0.33}$ x the basal value of 1) led to a sloppier landscape along with increase in phase-locking (shown below).

This phenomenon is expected to remain true for other parameters based on prior work by Abraham et al. (doi: 10.1038/msb.2010.92) that shows oscillators with a slow approach to the limit cycle (slow relaxation times) can be induced to phase-lock more easily. Slow relaxation times indicate smaller velocity vectors perpendicular to the limit cycle, which in turn implies a less steep topology on the approach to the limit cycle on the energy landscape. We have clarified our analysis and expanded our discussion of this point in the main text (first paragraph of Discussion).

2. Could the authors clarify the value of parameter A in Fig. EV3B? Is it an exact value or a fold-change relative to the original value (30.5, as indicated in Table EV1)? Could the authors explain the rationale for selecting A to be 1.2~1.4? Are those values within the oscillatory regime?

The values of the parameter A in Fig. EV3B were 1.2 – 1.4x multipliers of the original value of A = 30.5. The figure panel has now been updated for clarification. The values were chosen to

capture the transition between the values of A shown in Fig. 3 from a rigid to a sloppy landscape for $A = 30.5$ and $A = 30.5 * 10^{0.22} = 30.5 * 1.66$. These values of A all result in oscillatory p53 for an initial low constant level of DSB and a stable fixed-point value for the high constant level. As the system transitions to a less rigid oscillatory landscape with the increase in A , the transient oscillations dampen more rapidly. (Updated Fig. EV3B)

3. The authors hypothesize that changes in p53 pulsing frequency, as a result of phase resetting, influence the downstream response (Figure 4). Could the authors elaborate and discuss how they deconvolute the specific effects of pulsing frequency from other dynamic features, such as amplitude, total signal (area under the curve), or additional temporal characteristics, on downstream target regulation?

In this manuscript, we chose to focus on the effects of driving p53 expression at altered frequencies since the effects of p53 pulse frequency on downstream target expression have been well characterized (Porter et al. 2016 Cell Systems PMID: 27135539; Hafner et al. 2017 Nat Struct Mol Biol, PMID: 28825732). The 2nd pulse amplitudes do not appear to vary significantly between the different dose intervals tested which corroborates the digital nature of p53 pulses under DSB reported previously (Geva-Zatorsky et al 2005). The reviewer does make an excellent point that other features such as total signal (area under the curve) are also being modulated by repeated dosing and thus, affect downstream gene expression; however, such effects are difficult to deconvolve in our analysis. We have therefore revised the text to be more precise in our language in describing the changes to p53 dynamics resulting from phase resetting that led to the observed alterations of target gene regulation and cell fate.

4. Could the authors clarify how escaper cells are defined in Figure 4? Is the classification solely based on molecular characteristics (e.g., p21 level and/or Cdk2 activity) or does it also consider cellular phenotypes, such as cell division events?

Escaper cells were defined solely on p21 and Cdk2 levels. Cell division was not considered here due to potential confounding that can occur from failed cell division events (*i.e.*, endoreduplication, Toettcher et al, 2009, PNAS, PMID: 19139404). We have clarified this point in the text.

5. To further enhance the significance of this study, we suggest the authors to expand the discussion on how different p53 phase resetting frequencies could generally impact the expression of its target genes with different mRNA half-lives, thereafter cellular phenotypes. For instance, as the authors have mentioned, genes with unstable mRNA may require higher p53 frequency to activate. In addition, to activate a gene with weaker p53-binding promoter may require lower frequency of p53, as p53 needs to maintain in a higher level for a longer time period to initiate the transcription. Furthermore, genes with longer mRNA half-life may behave closer to an integrator. Lower p53 frequencies could lead to a higher rate of accumulation of these genes.

We appreciate the reviewer's support in expanding our discussion of the scope of our findings. At the reviewer's suggestion, we have expanded our discussion in the text of these important points.

10th Feb 2025

Manuscript number: MSB-2023-12077RRR

Title: Pulsed stimuli enable p53 phase resetting to synchronize single cells and modulate cell-fate

Dear Dr Batchelor,

Thank you again for sending us your revised manuscript. We are now satisfied with the modifications made and I am pleased to inform you that your paper has been accepted for publication.

Yours sincerely,

Sincerely,

Poonam Bheda, PhD
Scientific Editor
Molecular Systems Biology
